# Decomposing past and future: Integrated information decomposition based on shared probability mass exclusions

Thomas F. Varley[1,2]*

**1** Department of Psychological and Brain Sciences, Indiana University Bloomington, Bloomington, IN, United States of America, **2** School of Informatics, Computing, and Engineering, Indiana University Bloomington, Bloomington, IN, United States of America

* tvarley@iu.edu

**Data Availability Statement:** The data are all available on the CRCNS data sharing portal and can be downloaded here: https://crcns.org/data-sets/hc/hc-8/about-hc-8.

## Abstract

A core feature of complex systems is that the interactions between elements in the present causally constrain their own futures, and the futures of other elements as the system evolves through time. To fully model all of these interactions (between elements, as well as ensembles of elements), it is possible to decompose the total information flowing from past to future into a set of non-overlapping temporal interactions that describe all the different modes by which information can be stored, transferred, or modified. To achieve this, I propose a novel information-theoretic measure of temporal dependency ($I_{\tau sx}$) based on the logic of local probability mass exclusions. This integrated information decomposition can reveal emergent and higher-order interactions within the dynamics of a system, as well as refining existing measures. To demonstrate the utility of this framework, I apply the decomposition to spontaneous spiking activity recorded from dissociated neural cultures of rat cerebral cortex to show how different modes of information processing are distributed over the system. Furthermore, being a localizable analysis, $I_{\tau sx}$ can provide insight into the computational structure of single moments. I explore the time-resolved computational structure of neuronal avalanches and find that different types of information atoms have distinct profiles over the course of an avalanche, with the majority of non-trivial information dynamics happening before the first half of the cascade is completed. These analyses allow us to move beyond the historical focus on single measures of dependency such as information transfer or information integration, and explore a panoply of different relationships between elements (and groups of elements) in complex systems.

## 1 Introduction

What does it mean for a complex system to have "structure," or even to be a "system" at all? Nature abounds with systems: almost every object, when examined closely enough, is actually a composite structure, comprised of many interacting components. The world is a dynamic congeries of complex interactions and relationships. It is those relationships that define the

**Funding:** T.F.V is supported by NSF-NRT grant 1735095, Interdisciplinary Training in Complex Networks and Systems at Indiana University Bloomington.

**Competing interests:** The authors have declared that no competing interests exist.

nature and structure of the systems of which they are a part. For a system to have "structure," its behaviour in the future must be some consequence of its behaviour in the past. When parts of the system interact, the states of individual elements, or ensembles of elements, constrain their own possible futures, the futures of those components they interact with, and ultimately, the future of the system as a whole. For example, a single neuron embedded in a neuronal network might fire at some time $t − \tau$: that firing, constrains its own future (albeit transiently) due to subsequent hyper-polarization and the refractory period. It also informs on the possible futures of all those post-synaptic neurons to which it was coupled: the probability that they will fire changes after one of their parents fires and so on. In particular cases, the firing of a single neuron (or just a few neurons) may radically constrain the future of the entire brain (for example, if it triggers an epileptic seizure).

The entire scientific endeavour is, in some sense, built on uncovering these dependencies and understanding their specifics. For a complex system $\mathbf{X}$, comprised of many interacting parts, it is possible to quantify the total degree to which its future can be predicted based on its past with the *excess entropy* [1]:

$$E(\mathbf{X}) = I(\mathbf{X}_{-\infty:t}; \mathbf{X}_{t:\infty}) \qquad (1)$$

Where $\mathbf{X}_{-\infty:t}$ corresponds to the joint state of every element in $\mathbf{X}$, at every time $t$, from the first moment up to time $t$. The second term, $\mathbf{X}_{t:\infty}$, indicates the joint state of every element at every time from $t$ to the infinite future (I adopt the Python-like notation from [2]). Accounting for extended periods of past and future can reveal dependencies of varying durations (e.g. distance-related delays in communication networks), however, in practice, there are practical problems associated with recording infinite data, so the full excess entropy is typically inaccessible. In the particular case of Markovian systems, however, the situation is considerably easier, as the excess entropy reduces to the mutual information between a moment and its immediate past (possibly incorporating a lag of $−\tau$ moments):

$$E'(\mathbf{X}) = I(\mathbf{X}_{-\tau}; \mathbf{X}_t) \qquad (2)$$

For example, consider a two element system with Markovian dynamics: $\mathbf{X} = \{X^1, X^2\}$ (following [3] I use superscripts to denote indexes and subscripts to denote time). We can compute the lag-$\tau$ excess entropy of $\mathbf{X}$ as a whole as:

$$E'(\mathbf{X}) = I(X^1_{-\tau}, X^2_{-\tau}; X^1_t, X^2_t) \qquad (3)$$

The excess entropy is an extremely coarse measure, aggregating all of the temporal statistical dependencies, at every scale, within a multivariate system into a single number. For a more "complete" understanding of the dependencies within a system, it would be useful to be able to decompose it into non-overlapping components that describe how particular elements (and ensembles of elements) constrain each other as the system evolves through time: for example, how does the state of $X^1$ at time $t − \tau$ constrain its own future? How does it constrain the future of $X^2$? Other, more exotic dependencies are also possible: for example, the joint state of $X^1_{-\tau}$ and $X^2_{-\tau}$ together may constrain the future of just $X^1_t$ (a phenomena sometimes referred to as "downward causation", which has been the subject of intense philosophical debate [4, 5]). There may be information about the future of $X^1_t$ that is redundantly disclosed by observing either $X^1_{-\tau}$ alone or $X^2_{-\tau}$ alone, and so on. How can all of these different dependencies be untangled?

One possible path forward comes from the field of information decomposition. Classically, information decomposition concerns itself with the question of how to best understand how different ensembles of *predictor* variables collectively disclose information about a single *target*

variable [6, 7]. Since the original introduction of the *partial information decomposition* (PID) framework by Williams and Beer in 2010, researchers in complex systems science, information theory, and theoretical neuroscience have collectively worked to deepen our understanding of multivariate information and higher-order dependencies. Recently, Mediano, Rosas, and other introduced a multi-target information decomposition (the *integrated information decomposition* (ΦID) [3, 8, 9] which extends the original PID framework to multiple targets, enabling a full decomposition of the excess entropy into non-overlapping, "atomic" components (which I will refer to as Φ*I*, or integrated information, atoms). Despite being a considerable leap forward in our understanding of multivariate, temporal information, like the original PID, the ΦID lacks a crucial element required for applications to real data: an operational definition of *multivariate redundancy*.

In this work, I propose such a redundancy function, termed $I_{\tau sx}$. Based on a recent single-target measure introduced by Makkeh et al., [10] our proposed measure generalizes the classic Shannon mutual information function to ensembles of multiple interacting elements that may redundantly disclose information about each-other. I begin by reviewing the classic, single-target PID, before generalizing to the ΦID. I then introduce the $I_{\tau sx}$ measure, and demonstrate its application in three constructed Markovian systems designed to display distinct dynamical differences, and finally empirical, neuronal spiking data recorded from dissociated cultures of mouse hippocampal cortex [11, 12]. I conclude by discussing the strengths and limitations of our measure, and the ΦID framework itself.

## 1.1 Partial information decomposition

**1.1.1 Intuition & the bivariate case.** Consider the the simple case with two *predictor* variables ($X^1$ and $X^2$) that jointly disclose information about a target variable $Y$. Basic information theory gives us the tools to asses how each $X^i$ individually informs on $Y$ (the marginal mutual informations, e.g. $I(X^i; Y)$), and how the joint state of both $X^1$ and $X^2$ together inform on $Y$: $I(X^1, X^2; Y)$. The relationship between the marginal and joint mutual informations is not always straight-forward, however: the sum of both marginal mutual informations can be *greater than* or *less than* the joint mutual information in various contexts. If $I(X^1; Y) + I(X^2; Y) > I(X^1, X^2; Y)$, then there must be some information about $Y$ that is *redundantly* present in both $X^1$ and $X^2$ individually, and so when the two marginal mutual informations are summed, that redundant information is "double counted." Conversely, if $I(X^1; Y) + I(X^2; Y) < I(X^1, X^2; Y)$, then there is information about $Y$ in the joint state of $X^1$ and $X^2$ that is only accessible when the two are considered together and not accessible by looking at any individual $X$. These comparisons of "wholes" to "parts" are only rough heuristics, however, as redundant and synergistic information can co-exist in a set of predictor variables [6]: the direction of the inequality only indicates whether synergistic or redundant information *dominates* the interaction.

The seminal contribution of Williams and Beer was to provide a mathematical framework that allowed for a complete decomposition of the joint mutual information into non-overlapping, additive "atoms" of information:

$$
\begin{aligned}
I(X^1, X^2; Y) \quad = \quad & Red(X^1, X^2; Y) \\
+ \quad & Unq(X^1; Y/X^2) \\
+ \quad & Unq(X^2; Y/X^1) \\
+ \quad & Syn(X^1, X^2; Y)
\end{aligned}
\tag{4}
$$

where $Red(X^1, X^2; Y)$ is the redundant information about $Y$ that could be learned by observing *either* $X^1$ or $X^2$ individually, $Unq(X^1; Y/X^2)$ is the information about $Y$ that is *uniquely*

disclosed by $X^1$ (in the context of $X^2$, a vice versa for the other unique atom), and $Syn(X^1, X^2; Y)$ is the synergistic information about $Y$ that can only be learned by observing $X^1$ *and* $X^2$ simultaneously. Furthermore, the "marginal mutual informations" can be broken down into the same atomic components:

$$
\begin{aligned}
I(X^1; Y) &= Red(X^1, X^2; Y) + Unq(X^1; Y/X^2) \\
I(X^2; Y) &= Red(X^1, X^2; Y) + Unq(X^2; Y/X^1)
\end{aligned}
\tag{5}
$$

The result (in the case of two predictor variables) is an under-determined system with three known values (the three mutual information terms) and four unknown values (each of the partial-information atoms). If any one atom can be determined, then the remaining three are resolved "for free." Classical information theory does not provide any specific functions for any of these terms [13], and consequently their development is an area of active, and on-going, research. It is most common to begin by defining a redundancy function [6], although approaches based on defining unique [14, 15] and synergistic information [16, 17] have also been proposed. Unfortunately, if the number of sources is greater than two, the resulting decompositions of the joint and marginal mutual informations are not so constrained and more advanced mathematical machinery is required to decompose the joint mutual information.

**1.1.2 The partial information lattice & möbius inversion.** For a collection of $N$ predictor variables $\mathbf{X} = \{X^1, \ldots, X^N\}$ jointly informing on a single target $Y$, we are interested in understanding how every $X_i \in \mathbf{X}$ (and ensembles of $X$s joint by the logical conjunction) disclose information about the target. This requires understanding all the ways that the elements of $\mathbf{X}$ can redundantly, uniquely, and synergistically share information. Williams and Beer showed that, given an measure of redundant (or shared) information between some collection of sources and the target (denoted $I_\cap(\cdot; Y)$ here), the "atomic" components of the joint mutual information are constrained into a partially ordered set called the *partial information lattice*. The derivation of the lattice will be briefly described below, but see Gutknecht et al., for a more complete discussion [7].

We begin by defining the set of *sources* that may disclose information about $Y$. This is given by the set of all subsets of $\mathbf{X}$ (excluding the empty set, denoted as $\mathcal{P}_1(\mathbf{X})$). Every (potentially multivariate) source can be thought of as an aggregated macro-variable, whose state is defined by the logical-AND operator over all of its constituent elements. For example, if our predictor variables are $X^1$, $X^2$ and $X^3$, then the collections of sources are:

$$
\mathcal{P}_1(X_1, X_2, X_3) = \left\{
\begin{array}{l}
\{X^1\}, \{X^2\}, \{X^3\}, \\
\{X^1 \wedge X^2\}, \{X^1 \wedge X^3\}, \{X^2 \wedge X^3\}, \\
\{X^1 \wedge X^2 \wedge X^3\}
\end{array}
\right\}
\tag{6}
$$

For some (potentially overlapping) collection of sources, $\mathbf{A}^1, \ldots, \mathbf{A}^k$, the redundancy function $I_\cap(\mathbf{A}^1, \ldots, \mathbf{A}^k; Y)$ quantifies the information about $Y$ that can be learned by observing $\mathbf{A}^1 \vee \ldots \vee \mathbf{A}^k$. The domain of the $I_\cap(\cdot; Y)$ is given by the set of all collections of sources such that no source is a subset of any other:

$$
\mathcal{A} = \{\boldsymbol{\alpha} \in \mathcal{P}_1(\mathcal{P}_1(\mathbf{X})) : \forall \mathbf{A}^i, \mathbf{A}^j \in \boldsymbol{\alpha}, \mathbf{A}^i \not\subset \mathbf{A}^j\}
\tag{7}
$$

This restriction means that that $\mathcal{A}$ is also partially ordered:

$$\forall \boldsymbol{\alpha}, \boldsymbol{\beta} \in \mathcal{A}, \boldsymbol{\alpha} \preceq \boldsymbol{\beta} \Leftrightarrow \forall \mathbf{B} \in \boldsymbol{\beta}, \exists \mathbf{A} \in \boldsymbol{\alpha} \text{ s.t. } \mathbf{A} \subseteq \mathbf{B} \tag{8}$$

The resulting lattice $\langle \mathcal{A}, \preceq, \rangle$ provides the scaffolding on which the full PID may be constructed. Every $\boldsymbol{\alpha} \in \mathcal{A}$ corresponds to a vertex on the lattice, and the ordering reveals a structure of increasingly synergistic information-sharing relationships. For a visualization of the partial information lattices for sets of two and three predictor variables, see Fig 1.

With the structure of the partial information lattice set and our as-yet-undefined redundancy function in place $I_\cap(\cdot; Y)$, it is possible to solve the PID for every $\boldsymbol{\alpha} \in \mathcal{A}$ using a Möbius inversion:

$$\Pi(\boldsymbol{\alpha}) = I_\cap(\mathbf{A}^1, \ldots, \mathbf{A}^{|\boldsymbol{\alpha}|}; Y) - \sum_{\boldsymbol{\beta} \prec \boldsymbol{\beta} \in \mathcal{A}} \Pi(\boldsymbol{\beta}) \tag{9}$$

By recursively defining the value of particular partial information atoms as the difference between the redundant information disclosed by a particular set of sources and the sum of all atoms lower on the lattice, the joint mutual information between an arbitrary number of predictor variables and a single target can be decomposed into non-overlapping components.

$$I(\mathbf{X}; Y) = \sum_{i=1}^{|\mathcal{A}|} \boldsymbol{\alpha}_i \tag{10}$$

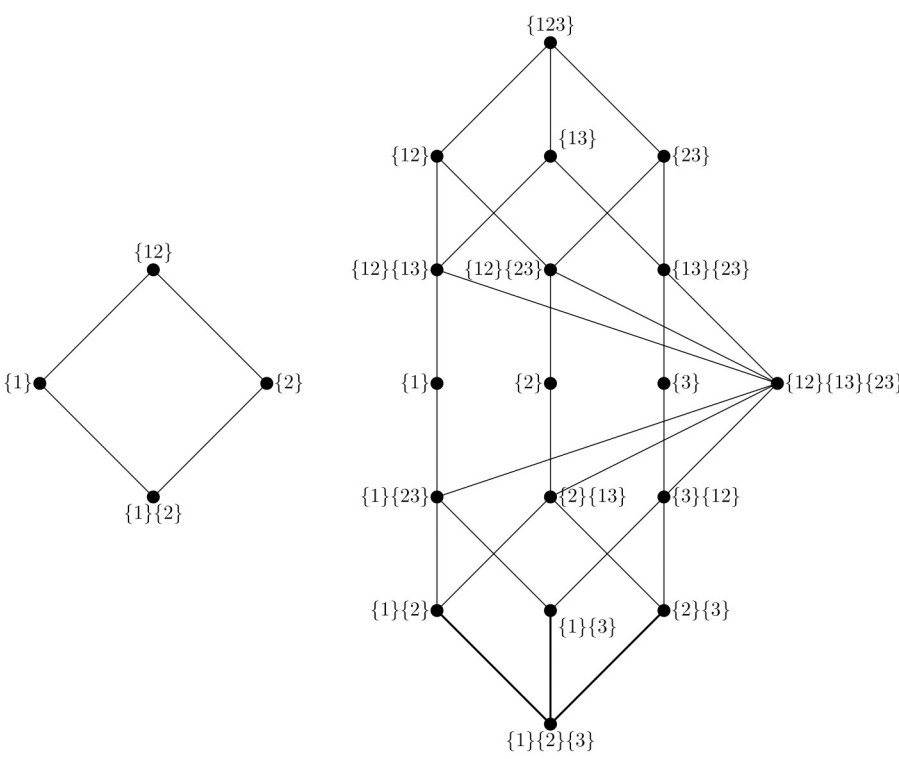

**Fig 1. Single target partial information lattices.** Examples of partial information lattices for the two simplest possible systems of multiple sources predicting a single target. On the left is the lattice for two predictor variables, and on the right is the lattice for three predictor variables. Following the notation introduced by Williams and Beer [6], sources are denoted just by index: for example {1}{2} is the information redundantly disclosed by $X^1$ or $X^2$, {1}{23} is the information disclosed by $X^1$ or ($X^2$ and $X^3$), etc.

## 1.2 Integrated information decomposition

With the basic PID defined, it is possible to do a partial examination of the excess entropy. For example, Varley and Hoel [18] decomposed the joint mutual information between all elements at time $t - \tau$ and the joint state of the whole system at time $t$: $I(X^1_{-\tau}, \ldots X^N_{-\tau}; \mathbf{X}_t)$. This method provides insights into how the states of particular elements (and ensembles of elements) collectively constrain the future of the whole system, but provides limited insights into how parts of the system constrain each-other, as the future state is aggregated into a single "whole." To partially address this issue, one could imagine doing a PID of the information that every element of the whole system at time $t - \tau$ discloses every set of elements at time $t$: decomposing $I(X^1_{-\tau}, \ldots X^N_{-\tau}; X^i_t), I(X^1_{-\tau}, \ldots X^N_{-\tau}; X^j_t), I(X^1_{-\tau}, \ldots X^N_{-\tau}; X^i_t, X^j_t)$, etc. While potentially illuminating, this approach is still limited by the fact that it does not readily allow for notions of "redundancy" and "synergy" between target elements. For example it might be natural to ask "what information synergistically disclosed by $X^i_{-\tau}$ and $X^j_{-\tau}$ about $X^m_t$ also applies to $X^n_t$ (i.e. is redundantly "copied" over both elements). Similarly, when decomposing $I(X^1_{-\tau}, \ldots X^N_{-\tau}; X^i_t, X^j_t)$, one might want to know what information about the joint state of $X^i_t$ and $X^j_t$ could *not* be learned by decomposing $I(X^1_{-\tau}, \ldots X^N_{-\tau}; X^i_t)$ or $I(X^1_{-\tau}, \ldots X^N_{-\tau}; X^j_t)$ alone (i.e. that information that is synergistically present in the joint state of $X^i_t$ and $X^j_t$ together). Achieving a complete decomposition of the excess entropy requires a generalization of the PID framework to account for redundancies and synergies in both the past *and future* of the system under study.

To address this, Mediano et al., [3, 8, 9, 19] recently introduced a generalization of the PID that allows the decomposition of multiple sources onto multiple targets. Called the *integrated information decomposition* (ΦID), this decomposition allows for a complete decomposition of the excess entropy.

The integrated information decomposition begins by defining a *product lattice* $\mathcal{A}^2 = \mathcal{A} \times \mathcal{A}$ (where $\mathcal{A}$ is the single-target redundancy lattice derived above), for which each vertex in $\mathcal{A}^2$ is defined by an ordered pair $\boldsymbol{\alpha} \to \boldsymbol{\beta}$, with $\boldsymbol{\alpha}, \boldsymbol{\beta} \in \mathcal{A}$. In the case of a temporal process, $\boldsymbol{\alpha}$ refers to a particular collection of sources observed at time $t - \tau$ that disclose information about $\boldsymbol{\beta}$, a collection of sources observed at time $t$.

As with the single-target partial information lattice, the product lattice is a partially ordered set, with:

$$\boldsymbol{\alpha} \to \boldsymbol{\beta} \preceq \boldsymbol{\alpha}' \to \boldsymbol{\beta}' \Leftrightarrow \boldsymbol{\alpha} \preceq \boldsymbol{\alpha}', \boldsymbol{\beta} \preceq \boldsymbol{\beta}' \tag{11}$$

The integrated information lattice can be similarly solved via Möbius inversion, given a suitable temporal redundancy function $I^{\boldsymbol{\alpha} \to \boldsymbol{\beta}}_\cap$. For a visualization of the integrated information lattice for the case of two sources and two targets, see Fig 2.

The ΦID framework deviates from the PID framework in one key way. In the original formulation by Williams and Beer, the lattice (which motivates the Möbius inversion) is derived from the axiomatic properties of the proposed redundancy measure. While there has never been universal agreement on the specific definition of "redundancy", any function that satisfies the original axioms can be shown to induce the lattice: it follows from the definition of redundancy. In contrast, in the ΦID framework, the double-redundancy lattice is not derived from the properties of the $I^{\boldsymbol{\alpha} \to \boldsymbol{\beta}}_\cap$ function, but rather, imposed by the product of the "marginal" PI lattices. To address this, Mediano et al., imposed a compatibility constraint on any double-redundancy function [3, 8]. Given two (potentially, but not necessarily) multivariate) variables $\mathbf{X}, \mathbf{Y}$

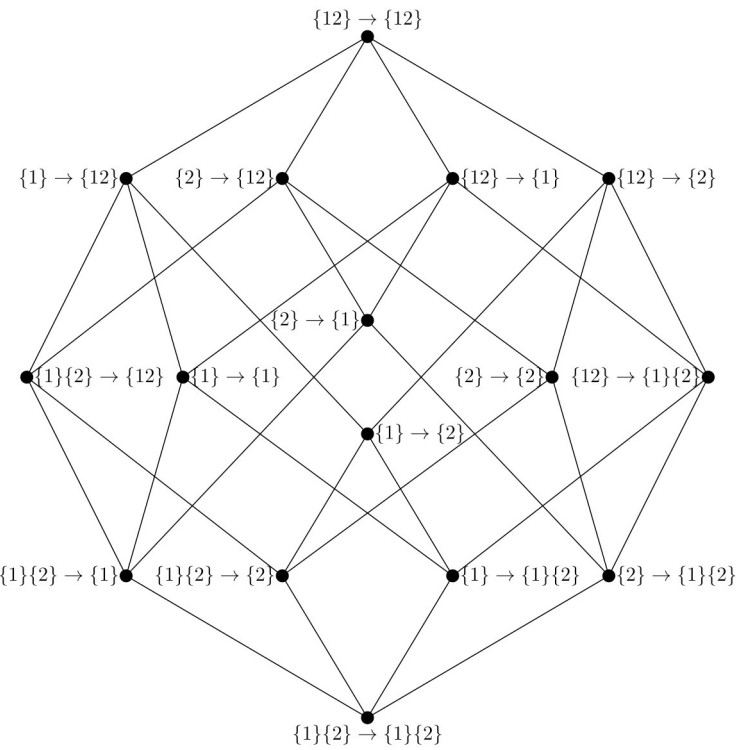

**Fig 2. The integration information lattice.** The integrated information lattice for a system $\mathbf{X} = \{X^1, X^2\}$. Every vertex of the lattice corresponds to a specific "conversion of information" that information in one mode at time $t - \tau$ can be transformed into at time $t$. For example, $\{1\}\{2\} \rightarrow \{1\}$ corresponds to information that is *redundantly* disclosed by $X^1$ and $X^2$ at time $t - \tau$ that is then only uniquely disclosed by $X^1$ at time $t$.

and two double-redundancy atoms $\boldsymbol{\alpha}, \boldsymbol{\beta} \in \mathcal{A}^2$

$$
I_\cap^{\alpha \rightarrow \beta} =
\begin{cases}
I_\cap(\mathbf{X}^{\alpha_1}, \ldots, \mathbf{X}^{\alpha_k}; \mathbf{Y}^{\beta_1}) & \Leftrightarrow |\mathbf{Y}| = 1 \\
I_\cap(\mathbf{Y}^{\beta_1}, \ldots, \mathbf{Y}^{\beta_j}; \mathbf{X}^{\alpha_1}) & \Leftrightarrow |\mathbf{X}| = 1 \\
I(\mathbf{X}; \mathbf{Y}) & \Leftrightarrow |\mathbf{X}| = |\mathbf{Y}| = 1
\end{cases}
$$

The compatibility axiom requires that, if one of the variables ($\mathbf{X}$ or $\mathbf{Y}$) is univariate, then the double redundancy function reduces to a classic, single-target redundancy function, and the $\Phi$ID reduces to the classic PID.

Mediano et al., also impose a partial-ordering criteria: if $\boldsymbol{\alpha} \rightarrow \boldsymbol{\beta} \preceq \boldsymbol{\alpha}' \rightarrow \boldsymbol{\beta}'$, then $I_\cap^{\alpha \rightarrow \beta} \leq I_\cap^{\alpha' \rightarrow \beta'}$. This ensures that the redundancy function induces the same partial ordering on atoms that the construction of the product lattice does, ensuring consistency between the scaffold and the function.

**1.2.1 Interpreting $\Phi$ID atoms.** The standard PID atoms are reasonably easy to interpret in terms of logical conjuctions and disjunctions of sources. In the case of the $\Phi$ID, the left-hand side of the integrated information atom remains the same (collections of sources that redundantly disclose information), but there is no longer a consistent target. Rather, there are again collections of sources that have their own redundant information sharing patterns. What, then, are they disclosing information *about*? I will discuss the answer in formal detail below, however, one proposed intuition is in the form of information dynamics. Information

dynamics proposes to break the different "modes" of information flow in complex systems down into discrete "types of computation" or "processing" [20]. Mediano et al. [3, 8], proposed the following intuitive taxonomy of integrated information atoms on the two-element lattice:

**Information Storage**: Information present in a particular configuration at time $t - \tau$ that remains in the same configuration at time $t$. In the case of the two-element system, these are: {1}{2} → {1}{2}, {1} → {1}, {2} → {2}, and {12} → {12}.

**Causal Decoupling**: The double-synergy term {12} → {12} has been given particular focus as a possible formal definition of "emergent dynamics" [9, 19, 21], as it refers to information that is persistently present in the whole, but none of the parts.

**Information Transfer**: Information present in a single element that "moves" to another single element: {1} → {2} and {2} → {1}. Not to be confused with the transfer entropy [22], which typically involves extended histories and itself conflates unique and synergistic modes of information sharing [23].

**Information Erasure**: Information that is initially present redundantly over multiple elements that is erased from one of the two: {1}{2} → {1} and {1}{2} → {2}.

**Information Copying**: Information that is initially present only a single element that is "duplicated" to be redundantly present in multiple elements. {1} → {1}{2} and {2} → {1}{2}.

**"Upward Causation"**: A somewhat less well-defined idea: when the state of single elements constrains the future state of the entire ensemble. {1}{2} → {12}, {1} → {1}{2}, and {2} → {1}{2}.

**"Downward Causation"**: A philosophically controversial concept, downward causation occurs when the synergistic joint state of the "whole" constrains the future of the individual parts. {12} → {1}{2}, {12} → {1}, and {12} → {2}

This taxonomy has only begun to be explored (for example see [24–26] for intriguing results related to macro-scale brain dynamics), and a rigorous formal understanding of the relevant mathematics may help deepen our understanding of these various (and in some cases, philosophically significant) phenomena.

## 2 Shared exclusions & (temporal) redundancy

A peculiar quirk of the PID and its derivatives is that, while it reveals the "structure" of multivariate information, it doesn't provide a direct means of calculating the specific values: it assumes the existence of a well-behaved redundancy measure and builds from there. Since the initial introduction by Williams and Beer, the number of different redundancy functions has proliferated (see [10, 13, 27–36]), although to date, no measure has achieved universal acceptance or satisfies every desiderata.

Being much newer, there has been less work on double redundancy functions: to date, only three have been used, most only once: a temporal minimum mutual information analysis [24, 26, 37], a measure based on the dependency lattice [9], and a generalization of the common change in surprisal measure [3]. While all analyses are informative, there is still room for deeper insights into the exact nature of temporal redundancy and how information *conversion* occurs between ensembles of variables. In this work, I generalize a recent redundancy function, the $I_{sx}$ measure first proposed by Makkeh et al., [10], to account for multiple targets which I term $I_{\tau sx}$. I selected $I_{sx}$ as my starting point for three reasons: the first is that it illuminates an elegant connection between multivariate information sharing and formal logic, second, because it does not require arbitrary thresholds (as in the case of $I_{ccs}$ [33]) nor non-

diffiable min/max functions (as in $I_{mmi}$ [31] and the closely related $I_{\pm}$ [34]). Third, it is *localizable*, returning values for every possible configuration, rather than being an expected value over the entire distribution. Below, I introduce the basics of local information theory (a key prerequisite for defining $I_{sx}$), before defining the redundancy function for single targets, and ultimately generalizing to multi-target information.

## 2.1 Local information theory

Thus far, I have been using the standard interpretation of mutual information as an average value over some distribution of configurations;

$$I(X; Y) := \mathbb{E}_{X,Y}\left[\log_2 \frac{P(y|x)}{P(y)}\right] \tag{12}$$

For any *specific* configuration, the *local mutual information* is defined as:

$$i(x; y) := \log_2 \frac{P(y|x)}{P(y)} \tag{13}$$

Unlike the expected mutual information, the local mutual information can be either positive or negative depending on whether $P(x|y)$ or $P(x)$ is the greater term. While the local mutual information is well-explored and has been previously used extensively to characterize "computation" in complex systems [20], it is only recently that a novel interpretive framework has emerged based on exclusions of probability mass. Finn and Lizier [38] showed that the sign and value of the local mutual information $i(x; y)$ can be understood as a function of the amount of probability mass from $P(X, Y)$ that is "ruled out" upon observing that $X = x$ and $Y = y$. For a very simple example, consider a system where one player rolls a fair die and another has to guess the value. Initially, the guesser is maximally uncertain, as all six outcomes are equiprobable. However, if they learn that the the number rolled was *even*, then they have *gained* information proportional to the total probability mass of all excluded possible outcomes. Formally, the local mutual information can be re-written in terms of probability mass exclusions as:

$$i(x; y) = \log_2 \frac{P(y) - P(y \cap \bar{x})}{1 - P(\bar{x})} - \log_2 P(y) \tag{14}$$

In this relationship, if $y$ is comparatively *more* likely after accounting for $x$, then $i(x; y) > 0$, and if it is less likely, then the value is negative.

## 2.2 Single-target redundancy based on shared exclusions (I$_{sx}$)

The classic, local mutual information is *bivariate*, quantifying the information shared between two variables. To construct a function that accounted for multiple sources redundantly disclosing information about a single target, Makkeh et al., leveraged a link between redundant information and logical implication [7, 10]. Briefly, given some set of logical statements $\psi^1, \ldots, \psi^k$, the information that is *redundantly* disclosed by all of them is the information learned if $\psi^1 = True$ OR $\psi^2 = True$ OR $\ldots \psi^k = True$. From this, they define a logical redundancy measure that induces the same lattice as the PID. The function provides a mapping between every $\alpha \in \mathcal{A}$ and a logical statement. For example, the atom {1}{2}{3} maps to the information disclosed if $\psi^1 \vee \psi^2 \vee \psi^3 = True$, and the atom {1}{2, 3} maps to the statement $\psi^1 \vee (\psi^2 \wedge \psi^3) = True$ and so on. The application to random variables is straightforward: given some set of source variables informing on a target $y$, the information about $y$ redundantly disclosed by all the sources

is the information that could be learned by observing $x^1$ alone OR $x^2$ alone . . . OR $x^k$ alone. The logic extends to more complicated atoms, such as {1}{2, 3}, which is the information about $y$ that would be learned by observing just $x^1$ alone OR the joint state of $x^2$ AND $x^3$ together.

As in the case of local mutual information, $i_{sx}$ defines "disclosing information" in terms of probability mass exclusions. For example, observing $X^1 = x^1 \lor (X^2 = x^2 \land X^3 = x^3)$ changes the probability of observing $Y = y$. As with the local mutual information, depending on how $P(y)$ changes, the value of $i_{sx}$ can be positive or negative.

More formally, consider a set of (potentially overlapping, potentially multivariate) sources $\mathbf{a}^1, \ldots, \mathbf{a}^k$ that collectively disclose information about a target $y$. The information redundantly shared between them can be defined as a function of the probability mass of $P(Y)$ that would excluded by observing $\mathbf{a}_1 \lor \ldots \lor \mathbf{a}^k$:

$$i_{sx}(\mathbf{a}^1, \quad \ldots, \mathbf{a}^k; y) :=$$

$$\log_2 \frac{P(y) - P(y \cap (\bar{\mathbf{a}}^1 \cap \ldots \cap \bar{\mathbf{a}}^k))}{1 - P(\bar{\mathbf{a}}^1 \cap \ldots \cap \bar{\mathbf{a}}^k)} - \log_2 P(y) \tag{15}$$

For the special case of only one source, it is clear that $i_{sx}(\mathbf{a}; y) = i(\mathbf{a}; y)$, which is itself just a regular joint mutual information: $i(x^{a^1}, \ldots, x^{a^{|a|}}; y)$. In this sense, $i_{sx}$ can be understood as generalizing the Shannon mutual information to account for ensembles of multiple sources that redundantly share information about $y$ [7].

Like the standard local mutual information $i_{sx}$ can return both positive and negative values (corresponding to informative and misinformative probability mass exclusions respectively). These two types of exclusion can be quantified by further decomposing $i_{sx}$ into two components:

$$i_{sx}^+(\mathbf{a}^1, \ldots, \mathbf{a}^k; y) \quad := \log_2 \frac{1}{P(\mathbf{a}^1 \cup \ldots \cup \mathbf{a}^k)} \tag{16}$$

$$i_{sx}^-(\mathbf{a}^1, \ldots, \mathbf{a}^k; y) \quad := \log_2 \frac{P(y)}{P(y \cap (\mathbf{a}^1 \cup \ldots \cup \mathbf{a}^k))} \tag{17}$$

$$i_{sx}(\mathbf{a}^1, \ldots, \mathbf{a}^k; y) \quad = i_{sx}^+(\mathbf{a}^1, \ldots, \mathbf{a}^k; y) - i_{sx}^-(\mathbf{a}^1, \ldots, \mathbf{a}^k; y) \tag{18}$$

In the context of a single-target PID, $i_{sx}^+$ and $i_{sx}^-$ are provably non-negative and satisfy the original desiderata proposed by Williams and Beer. The local redundant information measures can be aggregated into expected measures over the distribution of configurations in the same way as mutual information:

$$I_{sx} = \mathbb{E}_{\mathbf{A}^1, \ldots, \mathbf{A}^k, Y}[i_{sx}(\mathbf{a}^1, \ldots, \mathbf{a}^k; y)] \tag{19}$$

and likewise for the informative and misinformative functions.

## 2.3 Multi-target temporal redundancy based on shared exclusions ($I_{rsx}$)

We now have all the required machinery to introduce our local measure of temporal information decomposition: $I_{rsx}$. In the original $i_{sx}$ measure, the mutual information is understood as the relative increase or decrease in the probability $P(Y = y)$ after observing the configuration of some ensemble of sources. In $I_{rsx}$, the probability of the single target is replaced with the probability of observing $\mathbf{b}^1 \lor \ldots \lor \mathbf{b}^m$.

It is worth considering the intuition behind this change. Suppose $\mathbf{x} = \{x^1, x^2\}$ and $\mathbf{y} = \{y^1, y^2\}$. I am interested in what probability mass exclusions induced by $x^1$ OR $x^2$ are

consistent with either $y^1$ OR $y^2$. Said differently, what information that could be learned from either $x^1$ OR $x^2$ (i.e. is redundantly present in both of them) is true about *any* configuration consistent with $y^1$ OR $y^2$.

Formally:

$$i_{\tau sx}(\mathbf{a}^1, \ldots, \mathbf{a}^k; \quad \mathbf{b}^1, \ldots, \mathbf{b}^m) :=$$

$$\log_2 \frac{P(\mathbf{b}^1 \cup \ldots \cup \mathbf{b}^m)) - P((\mathbf{b}^1 \cup \ldots \cup \mathbf{b}^m) \cap (\bar{\mathbf{a}}^1 \cap \ldots \cap \bar{\mathbf{a}}^k))}{1 - P(\bar{\mathbf{a}}^1 \cap \ldots \cap \bar{\mathbf{a}}^k)} \tag{20}$$

$$-\log_2 P(\mathbf{b}^1 \cup \ldots \cup \mathbf{b}^m)$$

From here forward, I will denote ensembles of sources with $\boldsymbol{\alpha}, \boldsymbol{\beta}$, etc, for the purposes of notational compactness.

From the definition of $i_{\tau sx}$ it is clear that it follows the compatibility criteria proposed by Mediano et al., [3, 8]. When decomposing $i(\mathbf{x}, \mathbf{y})$ and $|\mathbf{y}| = 1$, then Eq 20 is equivalent to Eq 15 as the union of all sources ($\mathbf{b}^1 \cup \ldots \cup \mathbf{b}^m$) is equivalent to the single source $y$, and likewise for the condition where $|\mathbf{x}| = 1$. This shows that $i_{\tau sx}$ is consistent with the classic PID (inducing the standard single-target redundancy lattice). In the special case of single sources ($\boldsymbol{\alpha} = \{\mathbf{a}\}, \boldsymbol{\beta} = \{\mathbf{b}\}$), it is clear that $I_{\tau sx}(\mathbf{a}; \mathbf{b}) = i(\mathbf{a}; \mathbf{b})$ and so that $I_{\tau sx}$ completes the generalization of local mutual information begun by $i_{sx}$: $I_{\tau sx}$ is a full generalization of the mutual information to multiple sets of redundant sources and multiple sets of redundant targets.

**2.3.1 Multi-target redundancy & entropy decomposition.** The full $i_{\tau sx}^{\alpha \to \beta}$ function does not satisfy the partial ordering (monotonicity) criteria. Like $i_{sx}$, it can be negative or positive, depending on the structure of the dependency between the elements. The double-redundancy *can* be decomposed, however, into three *redundant entropy terms* that are partially ordered, and consistent with the integrated information lattice. These three redundant entropy terms induce three partial entropy decompositions [39–41]: two marginal decompositions on the classic redundancy lattice, and a joint decomposition on the product lattice.

The double redundancy function $i_{\tau sx}^{\alpha \to \beta}$ can be re-written in terms of sums and differences of union entropies. I can re-write Eq 20 in an equivalent form:

$$i_{\tau sx}(\boldsymbol{\alpha} \to \boldsymbol{\beta}) =$$

$$\log_2 \frac{1}{P(\mathbf{a}^1 \cup \ldots \cup \mathbf{a}^k)}$$

$$+ \log_2 \frac{1}{P(\mathbf{b}^1 \cup \ldots \cup \mathbf{b}^m)} \tag{21}$$

$$- \log_2 \frac{1}{P((\mathbf{a}^1 \cup \ldots \cup \mathbf{a}^k) \cap (\mathbf{b}^1 \cup \ldots \cup \mathbf{b}^m))}$$

For proof of equivalence, see Appendix A in S1 Appendix.

For some multivariate $\mathbf{x}$, recall that $i(\mathbf{x}; \mathbf{x}) = h(\mathbf{x})$. The joint entropy $h(\mathbf{x})$ can be decomposed by assessing how different combinations of *parts* (i.e. all $x^i \in \mathbf{x}$) redundantly and synergistically disclose information about the *whole* [10, 41]. The single-target $i_{sx}$ function can decompose $i$ $(x^1, \ldots, x^k; \mathbf{x})$ and will find that it is equal to $i^+(x^1, \ldots, x^k; \mathbf{x})$. There is no misinformative component (for proof, see Appendix B in S1 Appendix).

Intuitively, the redundant entropy function can be understood as quantifying how much uncertainty about $\mathbf{x}$ is resolved by learning $x^1 \vee \ldots \vee x^k$. This function was recently explored in

detail by Varley et al., [41] and denoted as $h_{sx}$ after [10]:

$$h_{sx}(\boldsymbol{\alpha}) = \log_2 \frac{1}{P(\mathbf{a}^1 \cup \ldots \cup \mathbf{a}^k)} \tag{22}$$

Which has been previously shown to satisfy the relevant Williams and Beer axioms locally [10]. I can then re-write Eq 20 as:

$$i_{\tau sx}(\boldsymbol{\alpha} \to \boldsymbol{\beta}) = h_{sx}(\boldsymbol{\alpha}) + h_{sx}(\boldsymbol{\beta}) - h_{sx}(\boldsymbol{\alpha} \cap \boldsymbol{\beta}) \tag{23}$$

This framing provides a different, but complementary perspective on $i_{\tau sx}$. The first term, $h_{sx}(\boldsymbol{\alpha})$ quantifies how uncertainty about the global joint state $(\mathbf{x}, \mathbf{y})$ is resolved by learning past states $\mathbf{a}^1$ OR $\ldots \mathbf{a}^k$ etc. Similarly, $h_{sx}(\boldsymbol{\beta})$ quantifies the uncertainty about $(\mathbf{x}, \mathbf{y})$ resolved by learning future states $\mathbf{b}^1$ OR $\ldots \mathbf{b}^m$.

The final term, $h_{sx}(\boldsymbol{\alpha} \cap \boldsymbol{\beta})$ is a little bit less straightforward, and reflects the structure of the double redundancy lattice and satisfies the required partial ordering imposed by Mediano et al., [8]. Makkeh et al., [10] showed that, if $\boldsymbol{\alpha} \preceq \boldsymbol{\alpha}'$ on the marginal (classic) redundancy lattice, then the set of configurations consistent with $\boldsymbol{\alpha}'$ is a subset of those configurations consistent with $\boldsymbol{\alpha}$. This ensures that $h_{sx}(\boldsymbol{\alpha}) \leq h_{sx}(\boldsymbol{\alpha}')$, and likewise for $\boldsymbol{\beta}$. The double redundant entropy term $h_{sx}(\boldsymbol{\alpha} \cap \boldsymbol{\beta})$ quantifies the probability of the *intersection* of the configurations consistent with $\boldsymbol{\alpha}$ AND $\boldsymbol{\beta}$. If $\boldsymbol{\alpha}' \subseteq \boldsymbol{\alpha}$ and $\boldsymbol{\beta}' \subseteq \boldsymbol{\beta}$, then $\boldsymbol{\alpha}' \cap \boldsymbol{\beta}' \subseteq \boldsymbol{\alpha} \cap \boldsymbol{\beta}$ and consequently, $h_{sx}(\boldsymbol{\alpha} \cap \boldsymbol{\beta})$ $\leq h_{sx}(\boldsymbol{\alpha}' \cap \boldsymbol{\beta}')$. For a worked example, see Appendix C in S1 Appendix. Note that, while $h_{sx}(\boldsymbol{\alpha} \cap \boldsymbol{\beta}) > 0$, it is not necessarily true that the associated partial entropy atoms are non-negative following the Mobius inversion.

From Eq 23 I can also construct the informative and misinformative probability mass exclusion formulation of $i_{\tau sx}$ equivalent to the informative and misinformative components of $i_{sx}$, however unlike in the single-target case, $i_{\tau sx}^-$ does not follow the partial ordering criteria that $h_{sx}$ does and is also not strictly non-negative.

**2.3.2 Interpreting $I_{\tau sx}$ in the $\Phi$ID.** Our analysis of $i_{\tau sx}$ and the decomposition into $h_{sx}(\boldsymbol{\alpha})$ + $h_{sx}(\boldsymbol{\beta})$ − $h_{sx}(\boldsymbol{\alpha} \cap \boldsymbol{\beta})$ has, thus far, been general, and could apply to any multivariate mutual information $i(\mathbf{x}; \mathbf{y})$. There is no temporal dynamic assumed. When decomposing $i(\mathbf{x}_{-\tau}; \mathbf{x}_t)$, the partial entropy terms can be understood as parcelling out the information contained in the *instantaneous* structure at time $t - \tau$ and time $t$, and the information in the *dynamics*: the transition from past the future. Each of the marginal partial entropy decompositions provides the entire, instantaneous structure at that moment. For example, when decomposing $h(\mathbf{x}_{-\tau})$ it is possible to extract all of the dependencies between the elements of $\mathbf{x}_{-\tau}$ (i.e the mutual informations $i(x_{-\tau}^i; x_{-\tau}^j)$, the conditional mutual information $i(x_{-\tau}^i; x_{-\tau}^j | x_{-\tau}^k)$ and so on [39, 41]), and likewise for $\mathbf{x}_t$. The sum, then of $h_{sx}(\boldsymbol{\alpha}_{-\tau}) + h_{sx}(\boldsymbol{\beta}_t)$ quantifies the total amount of information that could be learned about the transition $\mathbf{x}_{-\tau} \to \mathbf{x}_t$ without making any reference to the temporally extended dynamics of the system (note that if time were reversed, the sum of the entropy terms would be the same). It is the total "static" structure.

The dynamic structure is encoded in the last term $h_{sx}(\boldsymbol{\alpha}_{-\tau} \cap \boldsymbol{\beta}_t)$, which is the only term that incorporates information from the state-transition structure. This can help interpret those cases when $i_{\tau sx}(\boldsymbol{\alpha}_{-\tau} \cap \boldsymbol{\beta}_t) < 0$ bit. Negativity occurs when $h_{sx}(\boldsymbol{\alpha}_{-\tau}) + h_{sx}(\boldsymbol{\beta}_t) < h_{sx}(\boldsymbol{\alpha}_{-\tau} \cap \boldsymbol{\beta}_t)$. In plain language, this occurs when there is more information in the structure of the *transition* from $\mathbf{x}_{-\tau} \to \mathbf{x}_t$ then there is the instantaneous structures at time $t - \tau$ and $t$.

## 3 Results

In this paper, I have proposed a novel function of multi-target redundancy to be used as the foundation of an integrated information decomposition [3, 8]. Based on the logic of

information as exclusions of possible configurations [38], our proposed measure, $I_{rsx}$, general-izes the single-target redundancy measure first proposed by Makkeh et al., to enable the full decomposition of the excess entropy intrinsic to discrete dynamical processes. To demonstrate the measure in action, in the context of the ΦID, I will now explore some applications: the first three will be constructed systems designed to display markedly different dynamics (disinte-grated, integrated, and heterogeneous) to illustrate how different "types" of integration can be revealed by the decomposition. I will then examine spiking data from dissociated cultures made from rat brain tissue to demonstrate the insights that can be gained from both the expected, and local, integrated information decompositions.

## 3.1 Synthetic systems

Each of the three synthetic system is comprised of two, binary, elements that evolve through times according to different Markovian state-transition networks (visualized in Fig 3). Prior work on such simple, Boolean networks has shown that the space of even very small systems has a surprisingly rich distributions of redundant, unique, and synergistic effective informa-tion atoms [18]. Despite the simplicity of the synthetic systems under study here, they show-case how $I_{rsx}$ can reveal markedly different dynamic regimes. These systems were designed to show the two extreme behaviours of $\Phi^{WMS}$: the first system is totally dis-integrated and $\Phi^{WMS}$ = 0 bit (as the future of the whole can be perfectly predicted from the independent parts). The second system is completely integrated: the sum of the excess entropy of the parts is 0 bit, while the whole is has non-zero excess entropy. The third system is a heterogeneous combina-tion of integrated and dis-integrated dynamics. Considering the limiting cases of $\Phi^{WMS}$ can help build intuition about how the ΦID framework describes diverse dynamics. I hypothesized that the disintegrated system should, generally, not have much redundant temporal mutual information, as the elements are independent of each-other, and there should be little informa-tion transfer between individual elements. Similarly, in the case of the integrated system, I expected low redundancy, and a high degree of synergy (as the future of the whole can only be partially predicted by knowing the past of the whole).

**3.1.1 Disintegrated system.** The first system, $\mathbf{S}^D$ is a "disintegrated" system, in that each of the two dynamic elements is disconnected from the other: both predict *their own* futures with total determinism (the pattern is an oscillation $1 \to 0 \to 1 \to \ldots$), however there is no integration. Consequently, the excess entropy $I(\mathbf{S}^D_{t-1}; \mathbf{S}^D_t) = 2$ bit, and both individual excess entropies are each 1 bit: the "whole" is trivially reducible to the sum of its parts, since there's

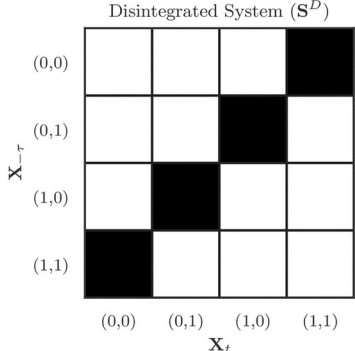
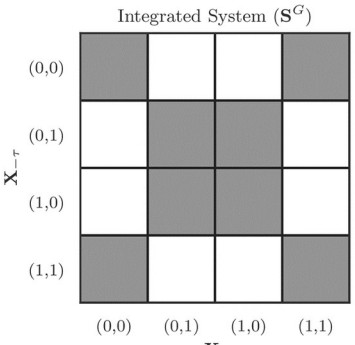
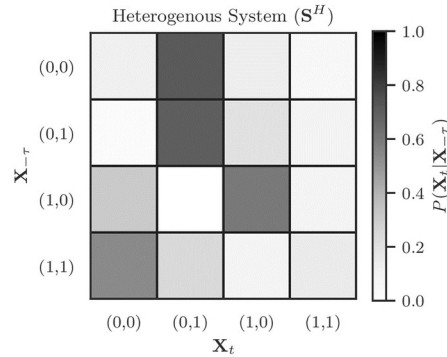

**Fig 3. Transition probability matrices for simple Boolean network systems.** On the left is a disintegrated system, where $E(\mathbf{X}) = E(X^1) + E(X^2) = 2bit$ (i.e. the whole is equal to the sum of the parts). In the middle is a highly integrated system, where $E(\mathbf{X}) = 1$ bit and both $E(X^i) = 0$ bit (i.e. the whole is greater than the sum of it's parts). On the right, a random system, combining a heterogenous mixture of integrated and segregated dynamics.

**Table 1. Table of integrated information atoms.** For each of the three Boolean systems, I performed the full integrated information decomposition, resulting in sixteen distinct ΦI atoms. The distinct global dynamics are reflected in the varying distributions of informative and misinformative information modes. I note that time-reversible systems (i.e. those where the probability of transitioning from state *i* to state *j* is the same as the reverse transition) have a more constrained and symmetrical structure than the heterogeneous system. Whether this is a universal fact about reversible versus irreversible dynamics remains an intriguing topic for future research.

| ΦI Atom | Disintegrated System | Integrated System | Heterogenous System |
|---|---|---|---|
| {1}{2} → {1}{2} | 0.415 | 0.152 | -0.021 |
| {1}{2} → {1}{1} | 0.0 | -0.152 | -0.006 |
| {1}{2} → {2}{1} | 0.0 | -0.152 | 0.099 |
| {1}{1} → {1}{2} | 0.0 | -0.152 | 0.008 |
| {1}{2} → {1}{2} | 0.0 | -0.152 | -0.005 |
| {1} → {2} | -0.415 | 0.152 | 0.077 |
| {2} → {1} | -0.415 | 0.152 | 0.055 |
| {1} → {1} | 0.585 | 0.152 | 0.037 |
| {2} → {2} | 0.585 | 0.152 | -0.072 |
| {1}{2} → {12} | 0.0 | -0.433 | 0.021 |
| {12} → {1}{2} | 0.0 | -0.433 | 0.049 |
| {1} → {12} | 0.415 | 0.433 | 0.032 |
| {2} → {12} | 0.415 | 0.433 | 0.005 |
| {12} → {1} | 0.415 | 0.433 | -0.008 |
| {12} → {2} | 0.415 | 0.433 | 0.075 |
| {12} → {12} | -0.415 | -0.018 | 0.078 |

no actual interaction between elements. For a visualization of the state-transition matrix, see Fig 3, left.

Decomposing the excess entropy using $I_{rsx}$ reveals several interesting relationships (for the full decomposition, see Table 1). As expected, the strongest information atoms are the element-wise information "storage" atoms: {1} → {1} and {2} → {2}. This is consistent with our intuition that the future of each element is *best* predicted by its own immediate past. The fact that the unique information storage atoms are also the largest single atoms is consistent with the idea that the most informative information dynamic about the whole system is the behaviour of the individual nodes considered individually. The informative interaction between the unique information and the synergistic information are also consistent with our intuitions about the disintegrated systems. For instance, the informative value of the "upward-causation" atom {1} → {12} reflects the fact that knowing the state of $X_{-\tau}^1$ also informs on the state of the whole $\mathbf{X}_t$ in the future: knowing $X_{-\tau}^1 = 0$ rules out any configuration of $\mathbf{X}_t$ where $X^t = 0$ (since the values of each $X^i$ oscillate). Likewise for the "downward-causation" atoms such as {12} → {1}: knowing the state of the *whole* at time $t - \tau$ constrains the individual parts at time $t$ (albeit, to a lesser extent than they constrain themselves).

The non-zero value of the double redundancy atom {1}{2} → {1}{2} is unexpected, although not inexplicable. Suppose that, at time $t - \tau$, $X_{-\tau}^1 = 0$ OR $X_{-\tau}^2 = 0$. Since both $X^i$ oscillate $0 \to 1 \to 0 \to 1\dots$, learning the state of either past variable is enough to rule out one possible joint future: $\mathbf{X}_t = (0, 0)$. Consequently, the union probability of all possible futures consistent with $X_t^1 \neq 0$ OR $X_t^2 \neq 0$ increases.

The final set of atoms worth exploring are the negative value for the information transfer terms (such as {1} → {2}). The temporal mutual information $I(X_{-\tau}^1; X_t^2) = 0$ bit, but the partial information atom is less than zero. Why? The answer is that computing the information transfer atom requires subtracting the sum of all the atoms that precede it on the lattice from the temporal mutual information. In this case, this is only the non-zero, double redundancy atom

{1}{2} → {1}{2}, which when subtracted off, produces a negative value (since the temporal mutual information is 0 bit).

**3.1.2 Integrated system.** The second system, $\mathbf{S}^G$ is an "integrated" system, in that there the whole system has 1 bit of excess entropy, but both elements have individual excesses entropies of 0 bit. This is accomplished using a parity check function: at every time step, the parity of the system is preserved, but the individual assignments are done randomly. For example, if $\mathbf{S}^G_{-\tau} = (1, 0)$, then $\mathbf{S}^G_t$ could equal $(0, 1)$ or $(1, 0)$ with equal probability (but never $(0, 0)$ or $(1, 1)$). For a visualization of the state-transition matrix, see Fig 3, center.

The first interesting finding is that, as with $\mathbf{S}^D$, the double-redundancy atom is unexpectedly positive. This occurs because, as before, learning $X^1_{-\tau}$ OR $X^2_{-\tau}$ is sufficient to inform on the future. Suppose that $X^1_{-\tau} = 0$ OR $X^2_{-\tau} = 0$. There are two configurations with an odd-parity consistent with those conditions ((0,1) and (1,0)), and only one configuration with even parity ((0,0)). This means that the future state will also be more likely to have an odd parity than an even one.

Due to the synergistic nature of the parity-check function, it is not possible to directly project from the parity of the joint state back down onto the states of the individual elements, which explains the negative values of the various information copy and erase atoms (such as {1} → {1}{2} and {1}{2} → {1}). The non-negative values of the various unique information storage and transfer atoms ({1} → {1} and {1} → {2}) occurs because the mutual informations between any pair of elements are all 0 bit (i.e. $I(X^1_{-\tau}; X^1_t) = 0$ bit, $I(X^1_{-\tau}; X^2_t) = 0$ bit, etc), however, the sum of all atoms lower down on the lattice is negative. For example, consider atom {1} → {1}. The excess entropy $E(X^1) = 0$ bit, however {1}{2} → {1}{2}+ {1}{2} → {1}+ {1} → {1}{2} = −0.152 bit, so the value of {1} → {1} works out to 0−(−0.152) bit: 0.152. The result can be admittedly difficult to interpret, although negative atoms in classic, single-target PIDs are also relatively widespread (see [10, 33, 34]). One possible interpretation, explored more in Section 3.1.4 is that these values reflect how the *context* provided by the dynamics of the whole system can influence our interpretation of dynamics of the individual parts. For example, when considered alone, there is no predictive information about the future of $X^1$ in it's own past, however, $X^1$'s evolution through time is not an autonomous process, but occurs *in the context of* $X^2$'s dynamics. Consequently, the kinds of inferences one can make about $X^1$ when considering the other elements of the system may be different from the kinds of inferences that could be made if $X^1$ were considered alone.

The higher-order, synergistic atoms show the same effect: since the redundancy-to-synergy atoms ({1}{2} → {12} and {12} → {1}{2}) are both negative, the upward and downward-causation atoms are positive, despite the fact that the relevant mutual informations are all zero bit.

**3.1.3 Heterogeneous system.** The final system was one with heterogeneous transitions, with probabilities drawn from a Gaussian distribution $\mathcal{N}(0, 1)$ (for details, see Varley & Hoel [18]). In contrast to the prior two systems, this system, $\mathbf{S}^H$ does not have an *a priori* fixed "type" of dynamic and was expected to display multiple types of information conversion. From the outset, I anticipated evidence of synergistic dynamics, as the excess entropy of the whole system was 0.422 bit, while each of the two elements had individual temporal mutual informations of 0.017 bit and 0.001 bit respectively, indicating a dynamic where the whole is much more predictive than the sum of its parts. For a visualization of the state-transition matrix, see Fig 3, right.

Consistent with expectations, $\mathbf{S}^H$ did not have the same regularity of information dynamics displayed by $\mathbf{S}^D$ and $\mathbf{S}^G$: for example, the atom {1}{2} → {1} was negative, indicating a misinformative relationship, while {1}{2} → {2} was positive and had a greater absolute value. Similarly, the conversion from redundant to synergistic information and vice versa both has

opposite signs, suggesting that this system simultaneously displays informative "downward causation", but misinformative "upward causation". In totality, there were more informative integrated information atoms than misinformative ones (a ratio of 11 to 5), showing that, despite the overall strongly synergistic nature of the system, unique information transfer and redundant information dynamics all co-existed together. This is consistent with previous work that found that these kinds of modified-Gaussian systems can display a wide range of information dynamics, at multiple scales [18].

**3.1.4 Interpreting negative local ΦI atoms.**   The three systems described above provide concrete toy models which can be used to build intuition about the phenomena of negative ΦI values. Consider the heterogeneous system $\mathbf{S}^H$: specifically the information that $S^2_{t-1}$ communicates to itself at time $t$ $(I(S^2_{t-1}; S^2_t))$. This value, sometimes called active information storage [20], is an expected mutual information and must be non-negative: in this case, it is 0.001 bit. If an observer were just observing the dynamics of $S^2$ (and ignoring $S^1$ entirely), their uncertainty about the future of $S^2$ would be reduced by observing its past. Despite this, the "stored" partial information {2}{2} is negative. How can this occur?

Our interpretation is that this mismatch is explained by the fact that the evolution of $S^2$ occurs *in the context of all other elements in the system*. Analysing $S^2$ on its own can be misleading, because its dynamics are informed by the states of the other elements. In this case, the lions share of the information is information that initially present in both atoms, and then erased from $S^1$: {1}{2} → {2}. So, a significant amount of the information that one observes in $I(S^2_{t-1}; S^2_t)$ is not *specific* to $S^2$ (at least not at first), but rather, emerges from the interaction with $S^1$.

In the case of these toy systems, there is no "mechanism" to be explored, however, this kind of distinction may be of value when analysing real-world systems. For example, a scientist studying the activity of neurons may observe a non-zero active information storage and propose to connect that to biophysical processes such as refractory periods [42], however, the ΦID analysis shows that this information may not actually reflect the specific dynamics of a single neuron, but rather, some contextual interaction between two, and the *actual* memory capacity of the neuron is altogether different (in this case, misinformative, rather than informative). This example is, at present, admittedly speculative, considerable work remains to understand how the outputs of the ΦID algorithm map onto complex, real-world dynamical process.

## 3.2 Analysis of dissociated neural culture data

To demonstrate how decomposition of the excess entropy using $I_{\tau sx}$ might be applied to empirical data, I analysed 31 dissociated cultures of rat hippocampal cortex. These preparations were made by resecting slices of embryonic rat cortex, and then culturing them to produce networks of living neuronal tissue [11]. After preparation of the cultures and a period of maturation, spontaneous spiking activity was then recorded on a 60 electrode array and spike-sorted to produce a time series of spikes for each putative neuron (for details, see the original manuscript presenting these data [12] and the Materials & methods Section).

Dissociated and organotypic cultures have been a highly productive model system for research into information dynamics and "computation" in biological systems: for example, see studies of the relationship between criticality and information-theoretic complexity [12, 43], network structure and synergy [44–46], changes to computational structure during maturation and development [47–49], and the topology of effective networks [50–52]. In many respects, they are a natural fit for these kinds of information theoretic analyses: neuronal activity is naturally discrete (in the form of action potentials which can be represented with binary states), the neuron is a well-defined "unit" (a single cell), and the communication channels between

units is well-understood at the mechanistic level (neurons communicate over synapses via the release of neurotransmitters), as are the general causal effects of interaction (neurons can be inhibitory or excitatory, a relationship easily expressible in terms of Bayesian prior and posterior probabilities [53]).

In this study, I demonstrate the utility of $I_{\tau sx}$ as both an expected and localizable measure of information-sharing by examining the pairwise relationships between neurons. Our particular focus is on avalanches of high-firing activity, which are typical of neural systems and systems poised near a critical phase transition in general. While the question of criticality in the brain is a complex question (for review, see [54, 55], and for a dissenting view, see [56]), it is an empirical fact that spontaneous activity in cortical networks displays avalanche dynamics of widely varying lengths (typically modeled as following a power-law, or other heavy-tailed distribution [57]). While the existence of such avalanches is extremely well-documented, and their genesis the subject of intensive modeling work, it is still unclear what, if any, role they play in cortical computations. Varley et al., hypothesized that they may play an integrative role after finding that loss of consciousness via the anaesthetic propfol caused pronounced collapse of large-scale avalanche structure [58], however, such hypotheses remain highly speculative in the absence of a formal framework for understanding localizable computation. I propose that the ΦID framework, coupled with the intrinsically local nature of $I_{\tau sx}$ solves that problem.

**3.2.1 Distributions of average ΦID atoms.**　For each of the 31 cultures, I calculated the lag-1 excess entropy for every pair of nodes in the network (restricting our analysis to consecutive bins within avalanches, as in [49]). If the expected excess entropy was significant at $\alpha = 10^{-6}$, (Bonferroni corrected), I went on to do the full integrated information decomposition. The result is, for every culture, across all pairs of nodes with significant excess entropy, I can compute sixteen distinct pairwise "integrated information matrices" (for visualization see Fig 4). For these expected values, I normalized each one by dividing it by its associated excess entropy to control for the variability in the overall amount of temporal information.

To explore the overall distribution of normalized information atoms, I aggregated over all cultures to create histograms of the various ΦID components (Fig 5). I found that the element-level information storage atoms ($\{x\} \rightarrow \{x\}$) had the overall highest average normalized value (0.417 ± 0.422), followed by the element-level information transfer atoms ($\{x\} \rightarrow \{y\}$, 0.097 ± 0.195). These results are consistent with our initial expectations: individual neurons are known to have a strong individual temporal dependence [42, 59], likely reflecting the refractory period following an action potential). Similarly, the high element-wise information transfer is consistent with the basic mode of communication between neurons being pairwise synaptic signaling. The other modes of information conversion, however, remain more mysterious: for example, the information copy and information erasure atoms ($\{x\} \rightarrow \{1\}\{2\}$ and $\{1\}\{2\} \rightarrow \{x\}$ respectively) both had values of 0.011 ± 0.0325, which is lower than the transfer atoms, but by less than an order of magnitude. Exactly what kind of biological process these modes correspond to is a promising area of future study. While every atom had particular pairs of neurons for which it was negative, at the aggregate level, every atom was, on average, greater than zero, including the higher-order measures, such as the double synergy ($\{12\} \rightarrow \{12\}$). These results show that spontaneous, on-going avalanche dynamics have a significant, element of consistently synergistic activity. For a complete set of correlations between all the atoms, see S1 Fig. I can also see that the information transfer atoms overall generally have the highest absolute values.

To compare the results of the integrated information decomposition to a more established measure of systemic complexity, I compared the distribution of normalized ΦID atoms to a measure of integrated information first proposed by Balduzzi & Tononi based on the

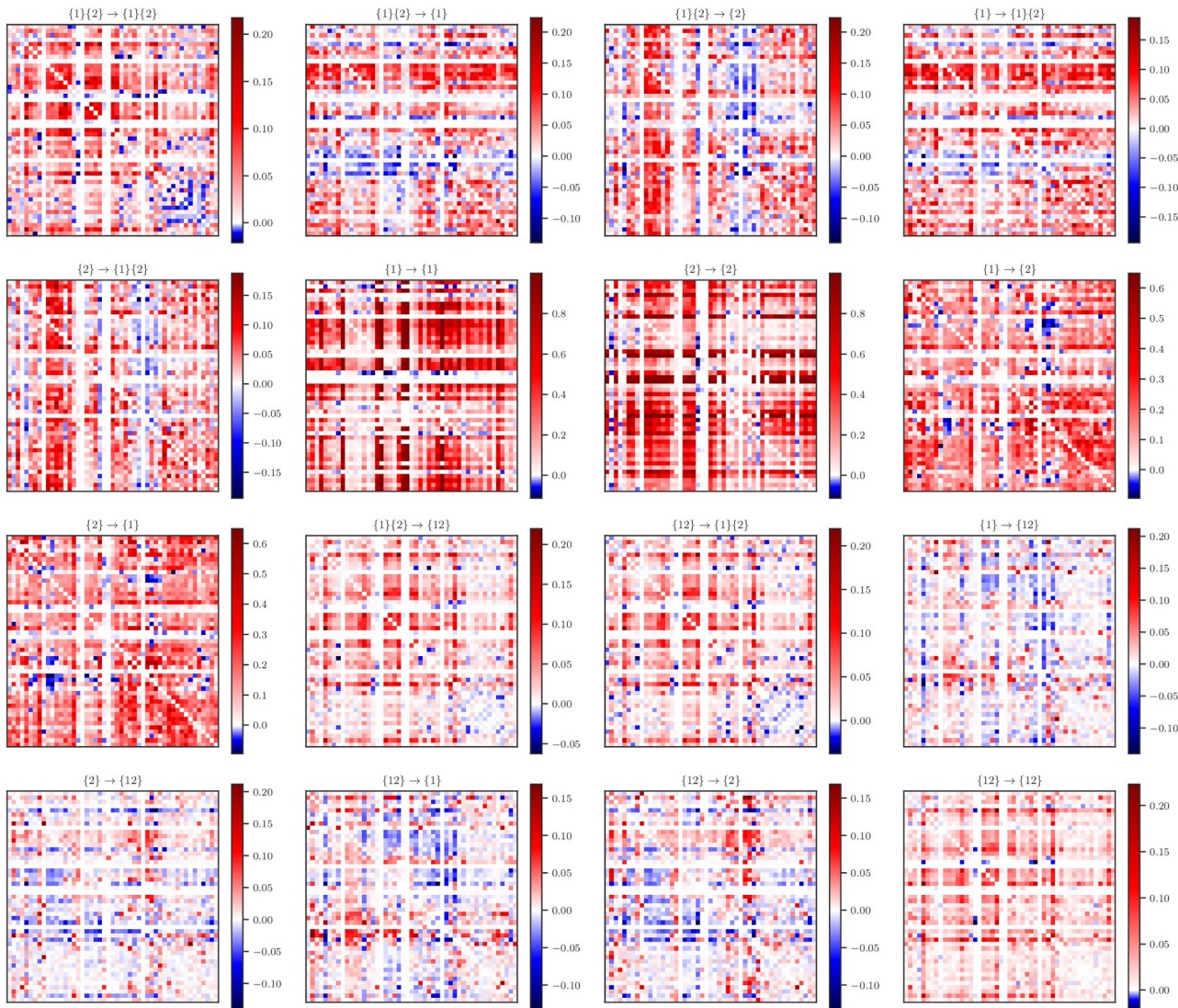

**Fig 4. Visualized normalized ΦI matrices for a single culture.** For a single culture (in this case with approximately one hundred individual neurons), one can construct sixteen different pairwise matrices, each one corresponding to a ΦI atom. This contrasts with more well-known measures of functional and effective connectivity, which produce one matrix per system, reflecting a single "kind" of statistical relationship (be it functional connectivity, effective connectivity, etc). Integrated information decomposition, on the other hand, provides multiple "kinds" of relationship at once, allowing a far more complete picture of computational dynamics. Here, the value of each atom is normalized by the total excess entropy.

difference between the total excess entropy and the sum of the two marginal excess entropies [60]:

$$\Phi^{WMS}(\mathbf{X}) = E(\mathbf{X}) - \sum_{i=0}^{|\mathbf{X}|} E(X^i) \tag{24}$$

Typically referred to as $\Phi^{WMS}$ ($WMS$ indicating "whole-minus-sum"), it is a useful measure of non-trivial systemic integration (see [37] for a recent exploration of $\Phi^{WMS}$ in a ΦID context). $\Phi^{WMS}$ has obvious parallels with the simple toy example of two predictors and a single target introduced in Section 1.1.1, with similar interpretations of the resulting sign (i.e. if $\Phi^{WMS} > 0$, then the system has synergistic dynamics only accessible when considering the

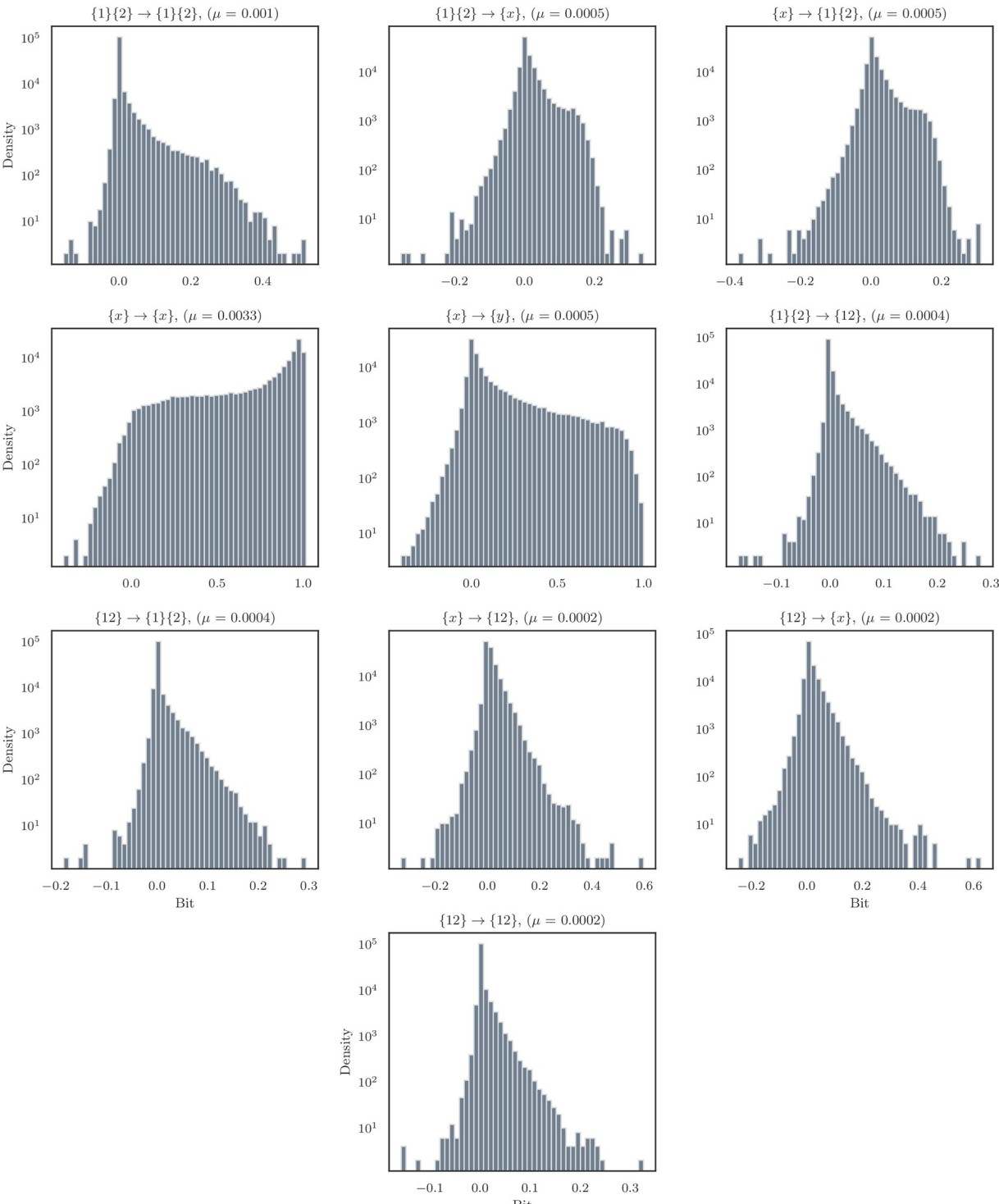

**Fig 5. Histograms of the normalized ΦI atoms across all cultures.** The distributions of the normalized atoms show marked differences, depending on the particular kinds of information conversion occurring. For example, the element-wise information storage atom as the highest mean value and is considerably biased towards informative, positive relationships, while other measures display a more symmmetric balance of informative and misinformative atoms (although all atoms displayed a bias towards informative relationships.

whole as opposed to the independent parts). As with the histograms, I aggregated over all significant pairs of neurons in all the cultures, and correlated each ones $\Phi^{WMS}$ against each of the normalized $\Phi$ID atoms. For visualization, see Fig 6.

Spearman correlation found that there was a very strong, negative correlation between $\Phi^{WMS}$ and the normalized information storage atoms ($\{x\} \rightarrow \{x\}$, $\rho = -0.8$, $p < 10^{-6}$, Bonferroni corrected). This is unsurprising, as information storage contributes to the marginal, within-element predictive information and contributes nothing to the higher-order interactions that comprise "integrated" information (consider the "disintegrated" toy model described above in Section 3.1). All other normalized $\Phi$I atoms were positively correlated with $\Phi^{WMS}$. The highest correlation was with the element-wise information transfer atoms ($\{x\} \rightarrow \{y\}$, $\rho = 0.57$, $p < 10^{-6}$, Bonferroni corrected). Since inter-element information transfer is a core element of systemic "integration", and considering the overall high prevalence of bivariate transfer in the data (see Fig 5), this result is unsurprising. As expected, the $\Phi$I atoms containing higher-order synergies were all positively correlated with $\Phi^{WMS}$, with the double-synergy term having one of the highest overall correlations ($\rho = 0.41$, p $< 10^{-6}$, Bonferroni corrected). This is consistent with the interpretation that $\Phi^{WMS}$ is an overall measure of total total systemic integration.

**3.2.2 Local $\Phi$ID analysis.** In addition to the average values of the integrated information atoms, the $I_{\tau sx}$ measure is localizable, allowing us to do a full, sixteen-atom decomposition for every moment in time, for every pair of neurons with significant excess entropy. I can leverage this property to perform a detailed analysis of the avalanches as temporally-extended objects qua themselves (rather than treating them as single units sampled from some heavy-tailed distribution). Across all pairs of neurons in all 31 cultures, I aggregated all avalanches of length $k > 4$, and if I observed at least 50 instances of avalanches of length $k$, I averaged them to create an "average profile." Prior work with dissociated culture data has shown that avalanche profiles tend to be scaled versions of one another [12] (and references therein), showing a characteristic growth and then collapse of activity over the duration (for a visualization of the average avalanche profiles, see Fig 7, Upper Left). For every moment in the avalanches, I computed the local excess entropy, and then performed the $\Phi$ID using the local $i_{\tau sx}$ to explore how the computational dynamics vary over the course of the avalanche. For a visualization of the profiles of the avalanches, the excess entropy, and all $\Phi$I atoms, see Fig 7. Local $\Phi$I atoms were not normalized, as the local excess entropy is a signed value, complicating the interpretation of a normalized value.

Upon visual inspection, it is clear that the various $\Phi$I atoms have distinct profiles: for example, the profiles of the element-wise information storage and transfer atoms are characteristically similar to the excess entropy profiles, with rapid increases to a peak followed by a heavy tail. In contrast the double-synergy profile has a noisier shape, appearing to drop towards misinformation at the end of the avalanche. To explore these profile differences in more detail, I directly compared the spiking activity profiles to their associated informational profiles. I began by computing the *cumulative* profile for each avalanche: in the cumulative avalanche, every moment is given as the sum of all previous moments, including the current one (analogous to a cumulative probability distribution). I then scaled each distribution by dividing it by the final, cumulative value, forcing all cumulative avalanches to terminate at 1. Finally, I filtered outlying cumulative avalanches that had unusually extreme deviations under the assumptions that they were contaminated by noise. By plotting the cumulative information atom avalanche distributions against the cumulative spiking avalanche distributions, it is possible to assess how the growth and collapse of information atoms differs from the change in spiking dynamics (see Fig 8). If the information atoms track the spiking activity perfectly, then the resulting curves will fall on the $y = x$ line. Deviations from the line of symmetry indicate a

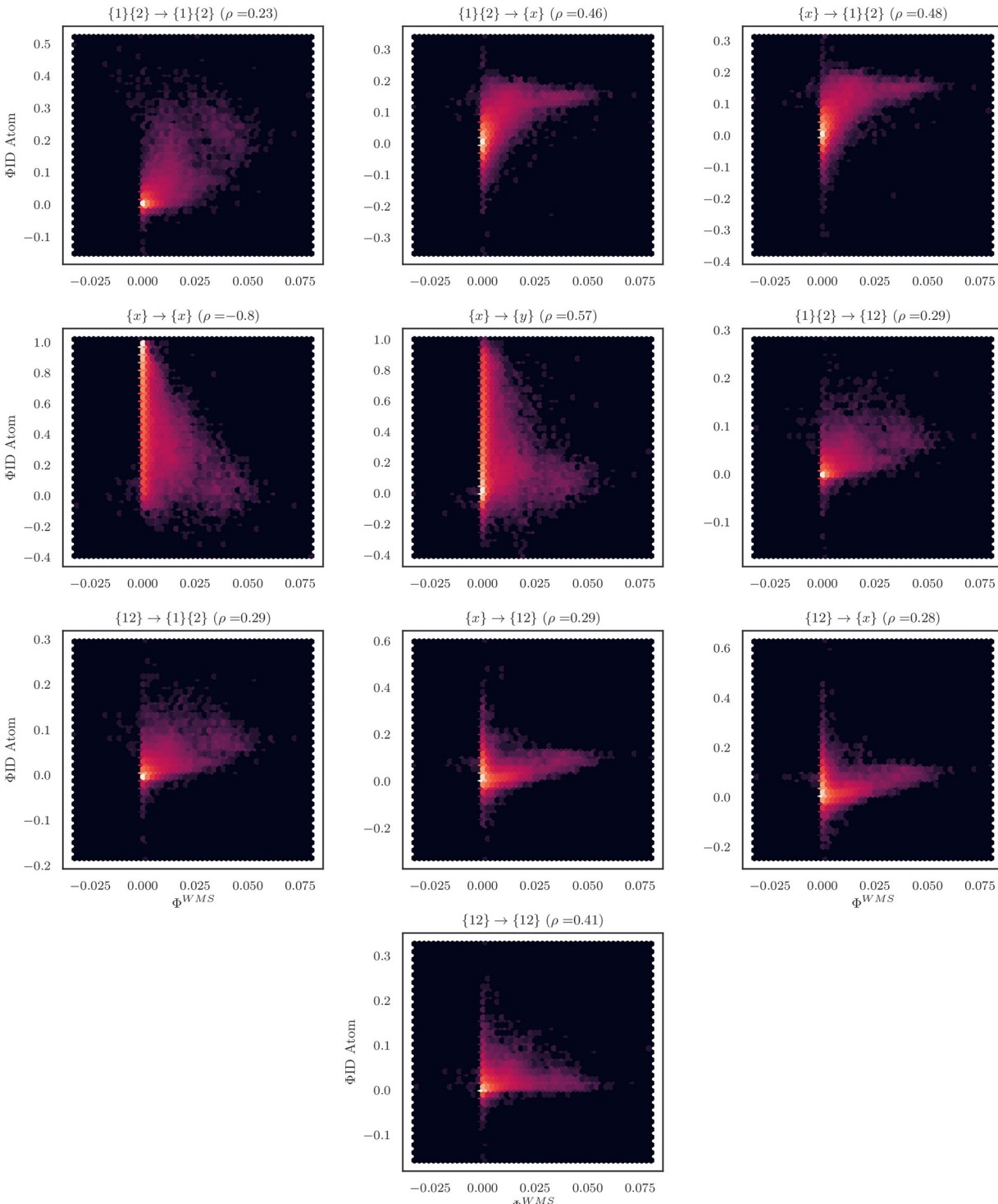

**Fig 6. $\Phi^{\mathbf{WMS}}$ vs. normalized $\Phi$I atoms.** The different normalized $\Phi$I atoms have varying degrees of correlation with the $\Phi^{WMS}$ measure of integrated information [60]. While most are generally positively correlated, the element-level information-storage atom is a dramatic outlier, with a highly significant negative correlation of -0.8. I believe this occurs because a high degree of information storage in single elements means that the future of the whole is *mostly* predictable from the individual parts. The more individual elements disclose about their own future, the less "integrated" information in the system.

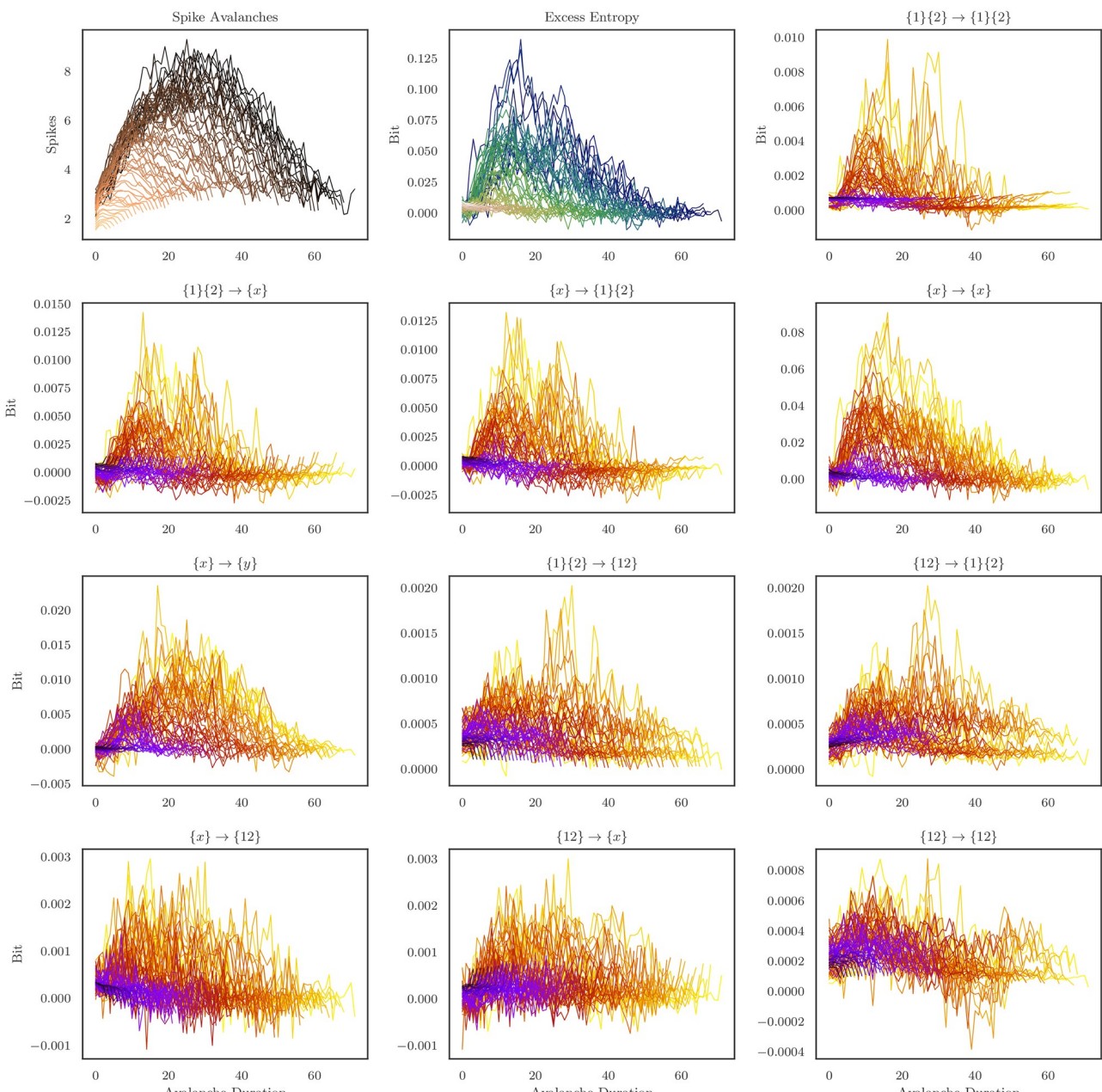

**Fig 7. Average avalanche profile plots for spiking activity.** Each curve is the average profile for avalanches of duration $k > 4$ bins, if at least fifty avalanches of that duration were observed across the all thirty-one cultures. On the upper-leftmost square, one can see the average profile for raw spiking activity (copper colormap). In the uppermost center plot, one can see the average profile for the local excess entropy (blue-green colormap), and for the rest of the plots, the remaining $\Phi$I atoms (violet-orange colormap). I can observe that different atoms have distinct characteristic profiles, some of which resemble the excess entropy more than others.

faster or slower accumulation of information than would be expected if it was perfectly corre-lated with spiking activity.

Visual inspection of the excess entropy cumulative profile reveals that avalanches are broadly-speaking informationally "front-heavy", the local excess entropy climbs much faster than spikes accumulate (as seen by the curve climbing *above* the $y = x$ line), and has almost

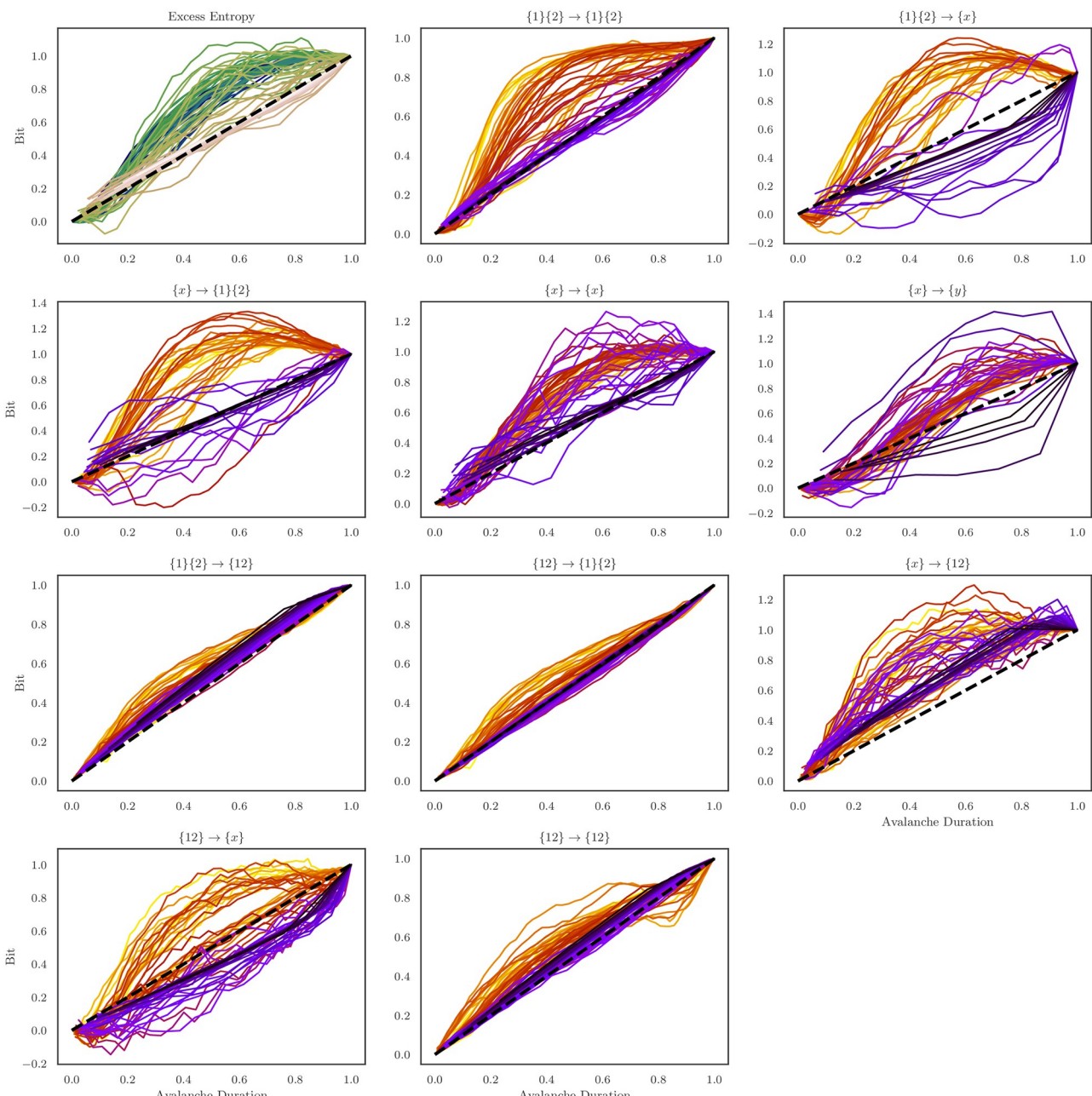

**Fig 8. Cumulative information avalanche profiles plotted against cumulative spiking avalanche profiles.** These plots allow us to assess how the density of information atoms varies over the duration of the avalanche *relative to the spiking activity that defines the avalanche*. The black dotted line indicates the $y = x$ line of symmetry: if the information density of an atom hugs that line, then the profiles of both the information and the spiking activity are the same. In many cases, the information profiles appear to dramatically diverge from the line of symmetry, indicating that avalanches are "informationally front-leaded", at least with respect to certain types of information integration.

entirely "saturated" before halfway through the avalanche. When considering avalanches of differing lengths, this front-heaviness appears to become more pronounced for larger avalanches (for small avalanches of length between 4 and 10, the normalized cumulative distribution curves hug the line of symmetry much more closely). This suggests that, while all spiking avalanche profiles may be roughly scaled versions of each-other, that scaling is not universal

when it comes to information content: larger avalanches have different information profiles than smaller ones.

The pattern displayed by the cumulative excess entropy profile is broadly mirrored by the individual $\Phi$I atoms, although is the considerable variation between them. For example, the synergy-to-redundancy atom {12} → {1}{2} (and it's mirror {1}{2} → {12}) both hug the line of symmetry much more closely. In contrast, the the cumulative double redundancy profiles and the cumulative information storage profiles track the cumulative excess entropy much more closely. Interestingly, the cumulative information copy and erasure profiles ({$x$} → {1}{2} and {1}{2} → {$x$}) both achieve a maximum value before the end of the avalanche and then drop down, indicating a transition from informative to misinformative dynamics towards the end of the activity period. The cumulative double-synergy profile shows one of the most intriguing patterns: for large avalanches, it appears to have an S-shaped profile, initially climbing rapidly during the avalanche, before dropping across the line of symmetry. The significance of such a dynamic is unclear, and this is a finding well worth revisiting and replicating in a future data set.

Another interesting type of variability between atoms is how the profile changes with avalanche duration. In the case of cumulative excess entropy, cumulative double-redundancy, and cumulaive information storage, small avalanches reliably hug the line of symmetry and it is the larger avalanches that display interesting deviations. However, this is not the only pattern: for example the "downward causation" atom ({12} → {$x$}) and the information erasure atoms both appear to display a kind of biphasic pattern: smaller avalanches (indicated by violet in Fig 8) run reliably *below* the line of symmetry, while large avalanches (indicated in orange) run above it.

From these results, it is clear that the $\Phi$ID framework, coupled with a localizable measure such as $I_{\tau sx}$ can provide a rich, novel approach to understanding ongoing neural activity and reveal patterns never before observed. For the purposes of this paper, I restricted myself largely to qualitative analysis of local integrated information dynamics: the results presented here will require ample replication and much deeper study to determine their significance.

## 4 Discussion

In this work, I have presented a novel information-theoretic measure, $I_{\tau sx}$, a generalization of the classic Shannon mutual information, that quantifies the redundant information shared between multiple sources and multiple targets. $I_{\tau sx}$ is motivated by the recently proposed Integrated Information Decomposition [3, 8], which generalizes the classic single-target Partial Information Decomposition [6, 7] to sets of multiple interacted sources and targets. Like all information decompositions, the $\Phi$ID is peculiar in that, while it reveals the *structure* of multivariate information, it lacks a crucial piece required to calculate numerical values from data. This is solved by providing $I_{\tau sx}$ as a redundancy function, with which the double redundancy lattice can be solved.

Here, the $\Phi$ID framework is used to decompose the excess entropy [1], which quantifies the total amount of statistical dependencies that constrains a systems evolution from past to future. Prior work [18] on using PID to decompose the excess entropy could reveal how the past states of individual components (and ensembles of components) constrain the future of the *whole* system, but provided no finer detail. Using the $\Phi$ID, it is possible to understand how elements constrain their own futures, the future of other elements, groups of elements or the whole system in much finer resolution. To demonstrate the utility of the $I_{\tau sx}$ measure, I first examined three small, completely specified toy models (each with its own enforced type of dynamic: integrated, disintegrated, or a mixture of the two) before moving on the empirical data recorded

from dissociated cultures of rat cortex. I showed that both the average and local versions of $I_{\tau sx}$ revealed rich information-dynamic structures in the data, including how different kinds of "neural computation" rise and fall as part of the bursty dynamics intrinsic to the nervous system.

A significant benefit of the ΦID framework is that is allows us to generalize different "kinds" of integration in a complex system such as the brain. Historically, information-theoretic approaches to integration have focused on single measures, such as integrated information theory's eponymous measure [60]. The information decomposition framework, however reveals a multitude of different ways that groups of neurons compute their next state. Recent, promising work using fMRI data has started to relate various ΦI atoms (particular the synergistic atoms) to macro-scale brain dynamics [24, 26], as well as different subcritical, critical, and supercritical dynamical regimes of various dynamical systems [37]. Given the wealth of data produced by modern neural recording methods, I am optimistic that there is a very wide world of possible applications of this framework.

While I have focused on the ΦID framework as a means of decomposing the excess entropy of ongoing, spontaneous neural dynamics in dissociated cultures, in principle the framework could apply to any data set with multiple, interacting predictor and predicted variables: the temporal dimension is not required. This opens up a wider range of applications of data analyses than is accessible to the classic PID—for example, Varley & Kaminski recently used the PID to asses how varying social identities (such as race and sex) jointly disclose information on single outcomes (such as income or health status) [61], however outcomes themselves are not independent and may contain interesting higher-order correlations within themselves. For example, how do the identities race and sex disclose information about income and health outcomes collectively? Generalizing to a ΦID framework may reveal many meaningful dependencies within social data, as well as many other fields where complex systems are studied.

## 4.1 Limitations

As currently formulated, the $I_{\tau sx}$ function is only well-defined for discrete random variables, a feature that it inherits from the original $I_{sx}$ measure [10]. Continuous generalization of $I_{sx}$ remains an area of active research [36] and it is assumed that a successful algorithm for $I_{sx}$ will also work for $I_{\tau sx}$. As it stands, the restriction to discrete random variables limits applicability. Prior work applying PID and ΦID to naturally continuous data such as fMRI or cardiac rhythms has been done using measures of redundancy that are well-defined for Gaussian distributions [24, 26, 37], although these measures have their own limitations, such as lacking the intuitive interpretation, being non-localizable, or requiring arbitrary thresholds or optimizations.

Even in the event that a successful generalization of $I_{\tau sx}$ is achieved, the PID and ΦID frameworks struggle to scale gracefully for all but the smallest systems. In the case of the PID, the number of atoms in the lattice of a system of size $k$ grows with the sequence of Dedekind numbers [7]: for a system with $k$ elements, the associated lattice has $D(k) - 2$ atoms. Given how fast the Dedekind sequence grows, a complete decomposition of almost any interesting natural system (which can have thousands, or millions of components) is impossible. The ΦID framework fares even worse, since there will be one temporal atom for every pair of partial information atoms in the associated PID lattice. The size of the ΦID lattice then grows with the mind-boggling square of the Dedekind numbers: $(D(k) - 2)^2$ (a five element system will have a ΦID lattice with 57,471,561 elements). Approximate heuristics such as the $\Phi^{WMS}$ measure, or more recently, the O-information [62–64] have been proposed as efficient, if imprecise, tools for recognizing the presence of higher-order dependencies in dynamical data,

however, there is still room for refinement. Another possibility might be to explore temporal PID (rather than $\Phi$ID) using a redundancy function equipped with a target chain rule [13]. The target chain rule generalizes the chain rule of mutual information to the PID: a redundancy function $i_\cap$ follows the rule if $i_\cap(\boldsymbol{\alpha}; y_1, y_2) = i_\cap(\boldsymbol{\alpha}; y_1) + i_\cap(\boldsymbol{\alpha}; y_2|y_1)$. Conveniently, the $i_{sx}$ measure satisfies the target chain rule, so a future avenue of research is to compare the results of $i_{\tau sx}$ with the chained $i_{sx}$.

The final limitation is that the the structure of the $\Phi$ID lattice, which allows for single sources to appear multiple times (e.g. $\{12\} \rightarrow \{x\}$ and $\{12\} \rightarrow \{12\}$ both incorporating the $\{12\}$ source) complicates the overall behavior of the redundancy functions. For example, the original $I_{sx}$ function has certain, provable properties (such as the global non-negativity of it's informative and misinformative components) that $I_{\tau sx}$ cannot adopt, since the structure of the lattice is different. This strong suggests that a return to the mathematical foundations of integrated information decomposition may be in order and new desiderata agreed on that may diverge from the single-target case.

## 5 Conclusions

In this work, I provide a redundancy function, $I_{\tau sx}$ that can be used to decompose the total information that flows from the past to the future through the "channel" of a multi-element, dynamic system. This framework, when applied to neural data reveals a rich repertoire of complex computational dynamics that can be temporally localized to the scale of individual moments in time. Based on the fundamental logic of information as exclusions of probability mass, $I_{\tau sx}$ generalizes the classic Shannon entropy and I anticipate that the work presented here will open new doors both in the specific fields of neuroscience as well as in complex systems science more generally.

## 6 Materials & methods

### 6.1 Dissociated culture preparation & recording

The details of the general process for the preparation of dissociated cultures can be found in [11]. Here I summarize the specific methodologies detailed in [12], who first introduced this dataset. Pregnant Sprague-Dawley rats (Harlan Laboratories) on Day 18 of gestation were euthanized via $CO_2$ and the embryos removed. Embryonic hippocampal tissue was ressected and dissociated en mass before being plated on a Multichannel Systems 60 electrode arrays ($8 \times 8$, 200 $\mu$m electrode spacing, 30 $\mu$m electrode diameter). Spontaneous activity was recording at 20,000 Hz for approximately 1 hour (for this analysis, all recordings longer than 60 minutes were terminated at that point). The resulting spikes were sorted with the wave_Clus algorithm [65] to infer individual neurons. Following spike sorting, the data were rebinned to 3ms bins (approximating the average inter-spike interval for the set of all 31 the recordings).

### 6.2 Mutual information calculation & significance testing

For every pair of neurons in a given culture, I calculated the mutual information between those two nodes at time $t$ and the same two nodes at time $t + 1$:

$$I(\mathbf{X}_t; \mathbf{X}_{t+1}) = H(\mathbf{X}_t) + H(\mathbf{X}_{t+1}) - H(\mathbf{X}_t, \mathbf{X}_{t+1}) \tag{25}$$

Where $H(\cdot)$ is the classical Shannon entropy function. I significance tested each pair against the analytic null distribution for discrete random variables with finite alphabets [66, 67], with an $\alpha = 10^{-6}$, followed by Bonferroni correction. The analytic null estimator allows for very efficient estimation of p-values, requiring minimal compute time (and reducing the associated

carbon costs associated with time-intensive high-performance computing). I used the implementation provided by JIDT [68], accessed via the IDTxl package [69] for its efficient Python interface.

### 6.3 Constructing toy Boolean networks

For the integrated and disintegrated example systems, the transition probabilities were worked out by hand from first principles. The heterogenous system was constructed based on the details provided in [18]. Briefly, a 4 × 4 transition probability matrix was initialized, and every entry $M_{ij}$ was drawn from a normal distribution with unit mean and variance. The absolute value was taken, and the out-going probabilities normalized to define a discrete probability distribution.

### 6.4 Excluding noisy cumulative avalanche profiles

To remove information avalanche profiles excessively contaminated by noise, I excluded any cumulative avalanche profiles that had an excursion of more than 1 bit away from the $y = x$ line or a total length greater than 2 bit. With these thresholds, I excluded on average 7.5 ± 8.18 avalanches for each ΦI atom. To see the full set of unfiltered cumulative avalanche plots, see S2 Fig.

### Supporting information

**S1 Fig. All pairwise correlations between normalized ΦI atoms.** Represented as two-dimensional log-probability density hexagonal histograms. The middle diagonal replicates the histograms seen in Fig 5. The correlations between various atoms are complex and not always trivial, or linear.
(PDF)

**S2 Fig. All cumulative avalanche plots without the filters.** Visual comparion with Fig 8 shows that the overall pattern can still be discerned despite the very noisy avalanches.
(PDF)

**S1 Appendix. Proofs & worked examples.** Various proofs and a worked example of computing $I_{\tau sx}$.
(PDF)

**S1 File.**
(ZIP)

### Acknowledgments

I would like to thank Dr. John Beggs & Dr. Olaf Sporns for mentorship and feedback on this project, and Ms. Maria Pope for feedback as well. I would also like to thank Dr. Abolfazl Alipour and Mr. Leandro Fosque for providing the dissociated culture data.

### Author Contributions

**Conceptualization:** Thomas F. Varley.

**Formal analysis:** Thomas F. Varley.

**Investigation:** Thomas F. Varley.

**Methodology:** Thomas F. Varley.

**Project administration:** Thomas F. Varley.

**Software:** Thomas F. Varley.

**Visualization:** Thomas F. Varley.

**Writing – original draft:** Thomas F. Varley.

**Writing – review & editing:** Thomas F. Varley.

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
