## [Decision Letter · Decision Letter 0]

26 Sep 2022

PONE-D-22-09166

Decomposing past and future: Integrated information decomposition based on shared probability mass exclusions

PLOS ONE

Dear Dr. Thomas Varely,

As before we thank you for your patience in the reviewing process. Also, thank you for submitting your manuscript to PLOS ONE. After careful consideration, we feel that it has merit but does not fully meet PLOS ONE’s publication criteria as it currently stands. Therefore, we invite you to submit a revised version of the manuscript that addresses the points raised during the review process.

Both referees are positive about the general ideas and direction of the paper. The first only suggest a minor revision a minor revision regarding the way "negative atoms" are considered within the paper. The second reviewer, however, has an  extremely detailed list of a number of points that should be addressed before the paper is accepted. 

We look forward to receiving your revised manuscript.

Kind regards,

Benjamin Z. Webb, Ph.D.

Academic Editor

PLOS ONE

ournal requirements:

2.For this single-authored manuscript, please replace "we" with "I".

4.Thank you for stating the following in the Acknowledgments Section of your manuscript:

“T.F.V is supported by NSF-NRT grant 1735095, Interdisciplinary Training in Complex Networks and Systems at Indiana University Bloomington. I would like to thank Dr. John Beggs & Dr. Olaf Sporns for mentorship and feedback on this project. I would also like to thank Dr. Abolfazl Alipour and Mr. Leandro Fosque for providing the dissociated culture data.”

5.In your Data Availability statement, you have not specified where the minimal data set underlying the results described in your manuscript can be found. PLOS defines a study's minimal data set as the underlying data used to reach the conclusions drawn in the manuscript and any additional data required to replicate the reported study findings in their entirety. All PLOS journals require that the minimal data set be made fully available. For more information about our data policy, please see http://journals.plos.org/plosone/s/data-availability.

Reviewers' comments:

Reviewer's Responses to Questions

**Comments to the Author**

1. Is the manuscript technically sound, and do the data support the conclusions?

Reviewer #1: Yes

Reviewer #2: Partly

2. Has the statistical analysis been performed appropriately and rigorously? 

Reviewer #1: Yes

Reviewer #2: N/A

3. Have the authors made all data underlying the findings in their manuscript fully available?

Reviewer #1: Yes

Reviewer #2: Yes

4. Is the manuscript presented in an intelligible fashion and written in standard English?

Reviewer #1: Yes

Reviewer #2: Yes

5. Review Comments to the Author

Reviewer #1: The article introduces a novel metric of double redundancy and evaluates it in various scenarios. The manuscript is very well written, the proposed metric has a number of desirable properties, and the presented analyses are very interesting. As I believe this is a good contribution to the field, I strongly suggest the manuscript to be accepted for publication.

My main suggestion is the following. It seems to me that this type of information decomposition do not consider negative atoms as a bug, but actually embrace them and provide an explanation for their meaning. If this is the case, I'd like to see that important feature highlighted more through the text. Additionally, it would be nice if this feature could be illustrated a bit more extensively in the analysis of the three synthetic models, where I would have liked to see a bit more of explanation of the meaning of various results.

Besides, it seems to me that this Phi-I-D could be readily extended for a local version, i.e. one could use these ideas to define an information decomposition that can be applied to individual datapoints. That could open an exciting line of research studying dynamics with these tools, which perhaps could be mentioned in the discussion? Also, it seems to me that such extension would be more natural than if one tries to do the same with other Phi-I-D decompositions?

Finally, I'd suggest to try to introduce the acronym "Phi-I" (which I guess stands for integrated information atom) in the introduction, just to make the reading easier.

Reviewer #2: * Summary

This paper builds upon the recently introduced Integrated Information Decomposition (phi-ID) from

Mediano et al. More specifically, it takes the proposed measure of redundant information I_sx from

Makkeh et al., which was originally introduced for the Partial Information Decomposition (PID), and

proposes a modified measure I_Tsx such that it is now compatible with Phi-ID. The paper discusses

some of the properties of the resulting decomposition, and applied it to several examples.

Have reviewed the paper, I have identified 10 significant issues that would need to be addressed

before the paper could be accepted. As such, I am recommending that this paper undergo major revisions.

* MAJOR ISSUES / COMMENTS

1. It is not clear to me if Phi-ID is meant purely for the analysis of timeseries, or can it be

applied to any random variables (as per PID). Some clarification on this point would be good.

2. You state that the desire to decompose the excess entropy into separate components associated

with the particular elements of the system is the main motivation behind this work, and you

describe how such a decomposition could perhaps be used to understand and quantify hitherto

weakly defined concepts such as downward and upward causation. This motivation is perfectly

reasonable and is of broad interest to the scientific community. You briefly discuss PID before

ruling it out because it is only applicable to a single target variable. However, based on what

you have written, I don't understand why this is so problematic.

In PID, the target is privileged in the sense we are decomposing the information that the source

variables provide about this particular variable. There is no restriction that requires the

target variable to be univariate, so in this regard, PID is not limited to a single target

variable. Of course, you are well aware of this since (as mentioned at the bottom of page 8) you

describe your own work (ref. [18]) which uses PID to decompose the information provided by each

state in the past about the entire system in the future. At top of page 9, you state that PID

provides no insights into how parts of the system constrain each other, as the future state is

aggregated into a single unitary whole. (Actually, you might want to reconsider describing a

multivariate variable as a unitary whole.) The problem I have is that clearly PID can also be

used to decompose the information provided by each state in the past about any particular state

in the future, or indeed about any collection of states in the future. If your application of

PID in ref. [18] provides insights into how the states of particular elements collectively

constrain the future of the whole system, then why would PID not be able to provide insights into

how the states of particular elements constrain particular parts of the system in the future? To

me, it seems that you just need to replace the target variable representing the future state of

the entire system with a target variable representing the future state of the particular part

that you are interested in. Hence, I do not understand how this is supposed to be a fundamental

limitation as you describe it. So, either I misunderstand the problem based on your the

description of the problem, in which case you should improve the description, or it is not quite

the limitation that you describe.

3. Following up on the prior point, it seems to me that your problem might perhaps be that

decomposing the information provided by each state in the past about a particular state in the

future would not strictly speaking be a decomposition of the excess entropy. However, I don't

really see how this is a problem. Nevertheless, if it is a problem, then it is worth noting that

certain approaches to information decomposition satisfy a chain rule for the target variable. In

effect, this means that there is consistency between the decomposition of the information

provided by each state in the past about the entire system in the future, and the separate

decompositions of the information provided by each state in the past about the particular states

in the future. That is to say, if the decomposition satisfying a target chain rule, then the

separate decompositions of the information provided by each state in the past about the future

state of each particular part in the future, must be consistent with the full decomposition of

the full decomposition of the information provided by each state in the past about the future

state of the entire system in the future, i.e. the decomposition of the excess entropy. In fact,

to me, this consistency seems like a very natural requirement. To summarise, I don't understand

the problem that you say is a limitation of PID, and furthermore, a PID with a target chain rule

seems like a much more important problem, as this would me that there is a consistency between

the decompositions of the information provided by the past state of the parts about future states

of the various parts,and the decomposition of the information provided by the past state of the

parts about future state of the entire system. Perhaps you could discuss the relevance of a

target chain rule to the problem of decomposition the future states of various parts of the

system versus the future state of the entire system.

4. PID is not yet a well-accepted methodology in information theory. This lack of acceptance is

largely down to a lack of a single, well-accepted measure of redundant information that is

compatible with the Williams and Beer Axioms. Nevertheless, if one accepts that the Williams and

Beer Axioms are reasonable properties for a measure of redundant information, then the derived

redundancy lattice cannot be dismissed. To be specific, Williams and Beer start with the Axioms

for I_cap, and then show how this axioms reduce the domain of I_cap to the set of all antichains

of the inclusion lattice, i.e. the set A(R) in their notation. They then use the axioms to show

that the elements of this set are partially ordered, and that these elements, together with the

partial order, form a lattice structure which they call the redundancy lattice. In other words,

if you accept the axioms, then you arrive at the redundancy lattice and this is not something

that is disputed. Furthermore, Williams and Beer then show how the axioms lead to a two nice

properties: specifically, that I_cap is nonnegative and an increasing function on the redundancy

lattice. In this regard, PID is theoretically sound and the fact that we do not yet have a

unique measure of redundant information is not problematic until you actually want to evaluate

the atoms in the lattice.

Similar to PID, Phi-ID is not generally accepted. However, unlike PID, Phi-ID does not have a

simple axiomatic basis. Reference [8] seems to provide two axioms, namely compatibility and

partial ordering. It seems to me that the compatibility axiom also contains the original

Williams and Beer axioms (although this is not very clearly stated), and so this axiom

effectively just states that we have the original Williams and Beer axioms. The second axiom,

i.e. partial ordering, however, is not so elegant. In effect, it is a requirement that this

double redundancy function satisfies a certain partial ordering, and there isn't much

justification provided for this axiom as far as I can tell. Contrast this to PID, where Williams

and Beer started with three very simple axioms and then subsequently showed that any function

which satisfies these axioms would necessarily be partially ordered. That is to say, in the

Williams and Beer approach, the lattice is derived from three simple properties of the redundancy

function, whereas in Phi-ID the lattice is effectively introduced as an axiom itself without much

justification. In fact, I am actually confused as to how exactly Reference [8] arrives at this

lattice structure from their axioms. It seems to come from Proposition 1 in Reference [8], but I

do not see any proof of this proposition in their paper.

Strictly speaking, this is a problem with Reference [8] rather than your paper. However, since

Reference [8] is not a published paper, this is also problematic for your paper. Since your

paper aims to build upon this work, I think it is important that you clarify how exactly the

double-redundancy lattice is obtained.

5. Following on from the prior point, you would also need to show that the double-redundancy

function that you define satisfies these axioms. If you can't (i) show that the double-redundancy

lattice can be derived from the axioms and then (ii) show that your double-redundancy function

satisfies these axioms, then you have no justification for later some of the later steps such as

the Moebius inversion. You will notice that almost all of the existing proposals for a measure

of redundancy for PID first start out by showing that the proposed measure satisfies the Williams

and Beer Axioms, or variants thereof.

6. Since you are building so directly upon i_sx, I think you need to improved the description of the

motivation behind its introduction. As it is, there isn't much beyond the sentence appearing

before equation (15). There is a great deal of discussion motivating i_sx in the original

papers, so it should be expanded since you are basically just extending this approach to make it

compatible with Phi-ID.

7. Similarly, when you introduce I_Tau-sx, you do not provide much of a description of the

motivation behind this particular definition. In particular, you should be justifying the

replacement that you make in your definition relative to the original definition of i_sx provided

in equation (15). Furthermore, is this change obvious? I am not saying that it is, but since the

original i_sx was proposed for PID rather than Phi-ID, then i_sx would clearly also need to be

modified. Perhaps the change you have made it a trivial change, or maybe it is more significant

in some way. Either case is fine, but you really aught to be discussing this change and

justifying why you have defined this generalisation in this particular way.

8. You are proposing a decomposition that is based on the local mutual information, which is a

non-negative function. The justification for doing this seems to come from reference [35] which

gets around this issue by decomposing the local entropy instead of the local mutual information,

i.e. the local mutual information is first decomposed into two non-negative local entropies, and

then each of these local entropies are decomposed separately. Thus, really you have two separate

information decompositions (i.e. two separate decompositions of the local entropy), and you are

combining the corresponding atoms of these two separate decompositions quantities to get

local-mutual-information-like quantities. In other words, you are not really decomposition the

local mutual information, but rather you are decomposing the two local entropies.

You present your decomposition as though it is a decomposition of the local mutual information,

and make it sound like the decomposition of the local entropies is an additional feature. To me,

it seems like it is actually a decomposition of the local entropy, and you are also combining

corresponding atoms of these two separate decompositions quantities to get

local-mutual-information-like quantities. This difference is important as the results from the

recombine atoms really should be seen as the results obtained from combining quantities from two

separate decompositions, rather than a decomposition in its own right. This problem is also

present in reference [10].

9. The synthetic systems that you use here in this paper: where do they come from? Are you

introducing them for the first time here, or are they well known within the field of IIT? Do we

know what the actual results should be or are their certain results which are seen as desirable

based on intuition?

In general, your description of obtain results does not give me the impression that the results

are well understood. For instance, when it comes to describing the result for the disintegrated

system, you describe the the element-wise storage atoms as being the "strongest" while the

pairwise transfer atoms are negative. You describe this latter result as being "consistent with

the notion of the system as a disintegrated structure", but I don't see how this is the case and

you do not provide any further clarification. You say that there are more positive

interactions(?) between individual elements and higher-order synergistic joint states than

anticipated. However, you never described or justified your anticipated result, so this is not a

helpful comment to make. The fact that you describe this result as "curious" does not fill me

with confidence that the results are indeed sensible or helpful. These results are all local

mutual informations, but as described in point 7, these values are really obtained from separate

decompositions of the two local entropies, so I suspect these results would be less mysterious if

you looked at these values.

Similarly, you are also a bit vague and uncertain about the results obtained from the integrated

system. Do we know what the results should be, or are is there some kind of intuitive result

that you would like to see? Describing some of the obtained results as a "surprise" when you are

considering synthetic examples that should have some definite result is quite problematic to

me. This could be fine if you provide an explanation for the surprising results, but I don't see

any justification that explains these results to me. This kind of description would be

absolutely fine if this was a well established method and you were applying it to some empirical

data from some system that is not well understood. It is troubling, however, when these are the

results that you obtain from applying this to synthetic examples which really should be

justifying the theoretical definition that you have introduced.

Regarding the heterogeneous system, I am not really sure what we are supposed to take away from

this example. You were actually a bit more explicit in stating your expectation for this

example, in that you said that you expected to see multiple types of information atoms present,

and indeed, this is the result that you obtained. However, what does it really tell us to learn

that a randomly generated example produces a dynamic which is associated with a small amount of

information associated with each of the atoms? Without further guidance, I am not sure what

exactly I should take away from this.

10. The empirical example using a dissociated neural culture: I don't want to be overly negative

here because I think the analysis of this kind of data is an interesting scientific endeavour,

and I think that information-theoretic methodologies are an appropriate way of potentially

understanding this kind of system. However, this entire section to me, is not helpful for a

paper that aims to propose a novel theoretical approach. It is interesting, but I think that

this work would be far more appropriate as a separate paper after this theoretical method has

been accepted in its own right.

The theoretical result should be interesting based on its own merits. You should propose the

theoretical measure, provide the motivation for the definition, and justify it using

well-understood examples. Typically, for this kind of proposal, you would typically use

synthetic examples (indeed, as you have aimed to do with your synthetic examples). This is

certainly true for all of the PID based approaches, although perhaps it is more tricky for

Phi-ID (see point 1).

The problem here is that the motivation for defining your measure is not very clear (see points

6 and 7), and I think there are some notable theoretical shortcomings (see points 4, 5 and 8).

Then the results the you obtain for the synthetic examples are not clear or convincing (see

point 9). With all of this in mind, I do not see what value the empirical example where there

is no well-established ground truth. If the results can't be fully explained for the synthetic

examples that are designed to exhibit certain features or characteristics, then I don't see how

we can trust the results when applied to real empirical data.

On the flip side, it is you paper, so I don't want to say that this analysis must be removed.

Rather, I am saying that it needs much better theoretical justification before we can trust the

results, or draw inference from the results when applied to empirical data. To reiterate, the

theoretical results should be interesting in there own right, and should stand on their own two

feet (to use an idiom). In my opinion, the level of detail required achieve this level of

theoretical rigour would result in a paper in its own right, and then this example could

potentially go in a separate empirical paper that uses the newly establish theoretical

framework. Whether or not you want to keep the empirical example in this paper, you must

improve the rigour of the theoretical results before this paper is suitable for publication.

6. PLOS authors have the option to publish the peer review history of their article (what does this mean?). If published, this will include your full peer review and any attached files.

Reviewer #1: No

Reviewer #2: No

---

## [Author Response · Author response to Decision Letter 0]

11 Nov 2022

\\documentclass{article}

\\usepackage[a4paper, total={6.5in, 9in}]{geometry}

\\usepackage{color}

\\usepackage{hyperref}

\\usepackage{amsmath}

\\newcommand{\\linespace}{\\color{white}X\\color{black}\\newline}

\\newcommand{\\response}[1]{\\linespace\\color{blue}#1\\color{black}\\linespace}

\\newcommand{\\atomspace}{\\color{white}\\{1\\}}

\\newcommand{\\atomspacet}{\\color{white}\\{1}

\\newcommand{\\atomspacett}{\\color{white}\\{1\\}\\}\\ }

\\newcommand{\\whiteminus}{\\color{white}-}

\\newcommand{\\boldalpha}{\\boldsymbol{\\alpha}}

\\newcommand{\\boldbeta}{\\boldsymbol{\\beta}}

\\begin{document}

 We would like to thank the two reviewers for taking the time to evaluate our manuscript and provide critical feedback. Below, our responses are noted in \\color{blue}blue font\\color{black}. 

 \\subsubsection*{Reviewer 1}

 The article introduces a novel metric of double redundancy and evaluates it in various scenarios. The manuscript is very well written, the proposed metric has a number of desirable properties, and the presented analyses are very interesting. As I believe this is a good contribution to the field, I strongly suggest the manuscript to be accepted for publication.

 My main suggestion is the following. It seems to me that this type of information decomposition do not consider negative atoms as a bug, but actually embrace them and provide an explanation for their meaning. If this is the case, I'd like to see that important feature highlighted more through the text. Additionally, it would be nice if this feature could be illustrated a bit more extensively in the analysis of the three synthetic models, where I would have liked to see a bit more of explanation of the meaning of various results.

 \\response{This is an excellent point, and one where there is a lot of space for discussion and further research. We have added the following:}

 \\color{blue}\\begin{quote}

 ``The three systems described above provide concrete toy models which we can use to build intuition about the phenomena of negative $\\Phi$I values. Consider the heterogeneous system $\\textbf{S}^{H}$: specifically the information that $S^{2}_{t-1}$ communicates to itself at time $t$ ($I(S^{2}_{t-1} ;S^{2}_{t})$). This value, sometimes called active information storage \\cite{lizier_local_2013}, is an expected mutual information and must be non-negative: in this case, it is 0.001 bit. If an observer were just observing the dynamics of $S^{2}$ (and ignoring $S^{1}$ entirely), their uncertainty about the future of $S^{2}$ would be reduced by observing its past. Despite this, the ``stored" partial information $\\{2\\}\\{2\\}$ is negative. How can this occur?

 Our interpretation is that this mismatch is explained by the fact that the evolution of $S^{2}$ occurs \\textit{in the context of all other elements in the system.} Analysing $S^{2}$ on its own can be misleading, because its dynamics are informed by the states of the other elements. In this case, the lions share of the information is information that initially present in both atoms, and then erased from $S^{1}$: $\\{1\\}\\{2\\}\\to\\{2\\}$. So, a significant amount of the information that one observes in $I(S^{2}_{t-1};S^{2}_t)$ is not \\textit{specific} to $S^{2}$ (at least not at first), but rather, emerges from the interaction with $S^{1}$.

 In the case of these toy systems, there is no ``mechanism" to be explored, however, this kind of distinction may be of value when analysing real-world systems. For example, a scientist studying the activity of neurons may observe a non-zero active information storage and propose to connect that to biophysical processes such as refractory periods \\cite{varley_information_2022}, however, the $\\Phi ID$ analysis shows that this information may not actually reflect the specific dynamics of a single neuron, but rather, some contextual interaction between two, and the \\textit{actual} memory capacity of the neuron is altogether different (in this case, misinformative, rather than informative). This example is, at present, admittedly speculative, considerable work remains to understand how the outputs of the $\\Phi ID$ algorithm map onto complex, real-world dynamical process."

 \\end{quote}

 Our hope is that this comment sheds light on how negative $\\Phi I$ atoms may be interpreted and reconciled with the non-negativity of Shannon mutual information.

 \\color{black}

 Besides, it seems to me that this Phi-I-D could be readily extended for a local version, i.e. one could use these ideas to define an information decomposition that can be applied to individual datapoints. That could open an exciting line of research studying dynamics with these tools, which perhaps could be mentioned in the discussion? Also, it seems to me that such extension would be more natural than if one tries to do the same with other Phi-I-D decompositions?

 \\response{The local $\\Phi ID$ analysis formed the lions-share of the avalanche analysis. Section B.2 (titled \\textit{Local $\\Phi ID$ Analysis}) describes a frame-wise decomposition over the course of neuronal avalanches.}

 Finally, I'd suggest to try to introduce the acronym "Phi-I" (which I guess stands for integrated information atom) in the introduction, just to make the reading easier.

 \\response{This is a good point and we have done so.}

 \\subsubsection*{Reviewer 2}

 This paper builds upon the recently introduced Integrated Information Decomposition (phi-ID) from

 Mediano et al. More specifically, it takes the proposed measure of redundant information $I_{sx}$ from

 Makkeh et al., which was originally introduced for the Partial Information Decomposition (PID), and

 proposes a modified measure $I_{\\tau sx}$ such that it is now compatible with Phi-ID. The paper discusses

 some of the properties of the resulting decomposition, and applied it to several examples.

 Have reviewed the paper, I have identified 10 significant issues that would need to be addressed

 before the paper could be accepted. As such, I am recommending that this paper undergo major revisions.

 * MAJOR ISSUES / COMMENTS

 1. It is not clear to me if Phi-ID is meant purely for the analysis of timeseries, or can it be

 applied to any random variables (as per PID). Some clarification on this point would be good.

 \\response{This is an excellent comment - the $\\Phi$ID is \\textbf{not} necessarily restricted to temporal dynamics, and could conceivably be applied to any data set with multiple predictor variables and multiple predicted variables (e.g. how might interacting demographic factors disclose information about interacting risk factors). We have expanded part of the Discussion to make this possibility more explicit:} \\begin{quote}

 \\color{blue}\\color{blue} ``...any data set with multiple, interacting predictor and predicted variables: the temporal dimension is not required. This opens up a wider range of applications of data analyses than is accessible to the classic PID - for example, Varley \\& Kaminski recently used the PID to asses how varying social identities (such as race and sex) jointly disclose information on single outcomes (such as income or health status) \\cite{varley_untangling_2022}, however outcomes themselves are not independent and may contain interesting higher-order correlations within themselves. For example, how do the identities race and sex disclose information about income and health outcomes collectively? Generalizing to a $\\Phi$ID framework may reveal many meaningful dependencies within social data, as well as many other fields where complex systems are studied."

 \\end{quote}

 Other, non-temporal applications are also certainly possible. \\color{black}

 2. You state that the desire to decompose the excess entropy into separate components associated with the particular elements of the system is the main motivation behind this work, and you describe how such a decomposition could perhaps be used to understand and quantify hitherto weakly defined concepts such as downward and upward causation. This motivation is perfectly reasonable and is of broad interest to the scientific community. You briefly discuss PID before ruling it out because it is only applicable to a single target variable. However, based on what you have written, I don't understand why this is so problematic.

 In PID, the target is privileged in the sense we are decomposing the information that the source variables provide about this particular variable. There is no restriction that requires the

 target variable to be univariate, so in this regard, PID is not limited to a single target variable. Of course, you are well aware of this since (as mentioned at the bottom of page 8) you describe your own work (ref. [18]) which uses PID to decompose the information provided by each state in the past about the entire system in the future. At top of page 9, you state that PID provides no insights into how parts of the system constrain each other, as the future state is aggregated into a single unitary whole. (Actually, you might want to reconsider describing a multivariate variable as a unitary whole.) The problem I have is that clearly PID can also be used to decompose the information provided by each state in the past about any particular state in the future, or indeed about any collection of states in the future. If your application of PID in ref. [18] provides insights into how the states of particular elements collectively constrain the future of the whole system, then why would PID not be able to provide insights into how the states of particular elements constrain particular parts of the system in the future? To me, it seems that you just need to replace the target variable representing the future state of the entire system with a target variable representing the future state of the particular part

 that you are interested in. Hence, I do not understand how this is supposed to be a fundamental limitation as you describe it. So, either I misunderstand the problem based on your the description of the problem, in which case you should improve the description, or it is not quite the limitation that you describe.

 \\response{The reviewer is completely correct in that you could conceivably do the PID of the past state of the whole system onto any particular subset of the system in the future (something like decomposing $I(\\textbf{X}_{-\\tau} ; X^1_t)$ or $I(\\textbf{X}_{-\\tau} ; X^2_t)$). We contend that this is a limited approach to decomposing the excess entropy, since it doesn't allow for the possibility of redundant (and by extension, synergistic) interactions among elements in the future. Without some well-defined notion of multi-target redundancy, it is difficult to understand what information disclosed by $I(\\textbf{X}_{-\\tau} ; X^1_t)$ and $I(\\textbf{X}_{-\\tau} ; X^2_t)$ is actually common to the future of both of them (i.e. that information that, in the $\\Phi ID$ formulation would belong to the $\\{1\\}\\to\\{1\\}\\{2\\}$ or $\\{2\\}\\to\\{1\\}\\{2\\}$ atoms). We have added the following:}\\color{blue}

 \\begin{quote}

 ``To partially address this issue, one could imagine doing a PID of the information that the joint state of the whole system at time $t-\\tau$ discloses every set of elements at time $t$: decomposing $I(X^1_{-\\tau},\\ldots X^N_{-\\tau} ; X^{i}_t)$, $I(X^1_{-\\tau},\\ldots X^N_{-\\tau} ; X^{j}_t)$, $I(X^1_{-\\tau},\\ldots X^N_{-\\tau} ; X^{i}_t, X^{j}_t)$, etc. While potentially illuminating, this approach is still limited by the fact that it does not readily allow for notions of ``redundancy" and ``synergy" between target elements. For example, when computing $I(X^1_{-\\tau},\\ldots X^N_{-\\tau} ; X^{i}_t)$, it might be natural to ask ``what information synergistically disclosed by $\\textbf{X}_{-\\tau}$ about $X^{i}_t$ also applies to $X^{j}_t$ (i.e. is redundantly ``copied" over both elements). Similarly, when decomposing $I(X^1_{-\\tau},\\ldots X^N_{-\\tau} ; X^{i}_t, X^{j}_t)$, one might want to know what information about the joint state of $X^{i}_t$ and $X^{j}_t$ could \\textit{not} be learned by decomposing $I(X^1_{-\\tau},\\ldots X^N_{-\\tau} ; X^{i}_t)$ or $I(X^1_{-\\tau},\\ldots X^N_{-\\tau} ; X^{j}_t)$ alone (i.e. that information that is synergistically present in the joint state of $X^i_t$ and $X^j_t$ together). To achieve a complete decomposition of the excess entropy, we need to generalize the PID framework to account for redundancies and synergies in both the past \\textit{and future} of the system under study."

 \\end{quote}

 It is also not at all clear that the result of doing the PID for every combination of elements in the future system could be parsimoniously recombined into the original excess entropy (i.e. it would not provide a true decomposition). For this reason, we think that the $\\Phi ID$ framework is a useful extension that cannot be easily replicated by mix-and-matching PIDs.\\color{black}\\newline

 3. Following up on the prior point, it seems to me that your problem might perhaps be that decomposing the information provided by each state in the past about a particular state in the future would not strictly speaking be a decomposition of the excess entropy. However, I don't really see how this is a problem. Nevertheless, if it is a problem, then it is worth noting that certain approaches to information decomposition satisfy a chain rule for the target variable. In effect, this means that there is consistency between the decomposition of the information provided by each state in the past about the entire system in the future, and the separate decompositions of the information provided by each state in the past about the particular states in the future. That is to say, if the decomposition satisfying a target chain rule, then the separate decompositions of the information provided by each state in the past about the future state of each particular part in the future, must be consistent with the full decomposition of the full decomposition of the information provided by each state in the past about the future state of the entire system in the future, i.e. the decomposition of the excess entropy. In fact, to me, this consistency seems like a very natural requirement. 

 To summarise, I don't understand the problem that you say is a limitation of PID, and furthermore, a PID with a target chain rule seems like a much more important problem, as this would me that there is a consistency between the decompositions of the information provided by the past state of the parts about future states of the various parts,and the decomposition of the information provided by the past state of the parts about future state of the entire system. Perhaps you could discuss the relevance of a target chain rule to the problem of decomposition the future states of various parts of the system versus the future state of the entire system.

 \\response{The possibility of a target chain rule as a solution to the problem of temporal PID is an intriguing one worthy of more extensive study, although I feel it is slightly beyond the scope of this article. We have added the following brief discussion, however I think this is a deep enough topic to warrant another study of it's own:}

 \\color{blue}

 \\begin{quote}

 ``Another possibility might be to explore temporal PID (rather than $\\Phi ID$) using a redundancy function equipped with a target chain rule \\cite{bertschinger_shared_2013}. The target chain rule generalizes the chain rule of mutual information to the PID: a redundancy function $i_{\\cap}$ follows the rule if $i_{\\cap}(\\boldsymbol{\\alpha};y_1,y_2)=i_{\\cap}(\\boldsymbol{\\alpha};y_1)+i_{\\cap}(\\boldsymbol{\\alpha};y_2|y_1)$. Conveniently, the $i_{sx}$ measure satisfies the target chain rule, so a future avenue of research is to compare the results of $i_{\\tau sx}$ with the chained $i_{sx}$."

 \\end{quote}

 With that said, I don't think a target chain rule-based decomposition of temporal MI would convey the same information that the $\\Phi$ ID framework does. My understanding is that conditioning can remove redundancies and reveal synergies, but cannot provide a complete decomposition (see Williams and Beer's discussion about co-information and conditional mutual information). Either way, I feel like this is better left for future work. \\color{black}

 4. PID is not yet a well-accepted methodology in information theory. This lack of acceptance is largely down to a lack of a single, well-accepted measure of redundant information that is compatible with the Williams and Beer Axioms. Nevertheless, if one accepts that the Williams and Beer Axioms are reasonable properties for a measure of redundant information, then the derived redundancy lattice cannot be dismissed. To be specific, Williams and Beer start with the Axioms for $I_{\\cap}$, and then show how this axioms reduce the domain of $I_{\\cap}$ to the set of all antichains of the inclusion lattice, i.e. the set A(R) in their notation. They then use the axioms to show

 that the elements of this set are partially ordered, and that these elements, together with the partial order, form a lattice structure which they call the redundancy lattice. In other words, if you accept the axioms, then you arrive at the redundancy lattice and this is not something that is disputed. Furthermore, Williams and Beer then show how the axioms lead to a two nice properties: specifically, that $I_{\\cap}$ is nonnegative and an increasing function on the redundancy

 lattice. In this regard, PID is theoretically sound and the fact that we do not yet have a unique measure of redundant information is not problematic until you actually want to evaluate the atoms in the lattice.

 Similar to PID, Phi-ID is not generally accepted. However, unlike PID, Phi-ID does not have a simple axiomatic basis. Reference [8] seems to provide two axioms, namely compatibility and partial ordering. It seems to me that the compatibility axiom also contains the original Williams and Beer axioms (although this is not very clearly stated), and so this axiom effectively just states that we have the original Williams and Beer axioms. The second axiom, i.e. partial ordering, however, is not so elegant. In effect, it is a requirement that this double redundancy function satisfies a certain partial ordering, and there isn't much justification provided for this axiom as far as I can tell. Contrast this to PID, where Williams

 and Beer started with three very simple axioms and then subsequently showed that any function which satisfies these axioms would necessarily be partially ordered. That is to say, in the Williams and Beer approach, the lattice is derived from three simple properties of the redundancy function, whereas in Phi-ID the lattice is effectively introduced as an axiom itself without much justification. In fact, I am actually confused as to how exactly Reference [8] arrives at this

 lattice structure from their axioms. It seems to come from Proposition 1 in Reference [8], but I do not see any proof of this proposition in their paper.

 Strictly speaking, this is a problem with Reference [8] rather than your paper. However, since Reference [8] is not a published paper, this is also problematic for your paper. Since your paper aims to build upon this work, I think it is important that you clarify how exactly the double-redundancy lattice is obtained.

 \\response{This is a very thoughtful and well-considered critique of the $\\Phi$ID framework that prompted a considerable amount of consideration on our part. There are a few points we would like to respond to: 

 The first is that there actually have been a number of published (i.e. peer reviewed) applications of the $\\Phi ID$ in a number of different journals. See:}

 \\color{blue}

 \\begin{quote}

 Rosas, F. E., Mediano, P. A. M., Jensen, H. J., Seth, A. K., Barrett, A. B., Carhart-Harris, R. L., \\& Bor, D. (2020). Reconciling emergences: An information-theoretic approach to identify causal emergence in multivariate data. PLOS Computational Biology, 16(12), Article 12. https://doi.org/10.1371/journal.pcbi.1008289

 Luppi, A. I., Mediano, P. A. M., Rosas, F. E., Harrison, D. J., Carhart-Harris, R. L., Bor, D., \\& Stamatakis, E. A. (2021). What it is like to be a bit: An integrated information decomposition account of emergent mental phenomena. Neuroscience of Consciousness, 2021(2). https://doi.org/10.1093/nc/niab027

 Luppi, A. I., Mediano, P. A. M., Rosas, F. E., Holland, N., Fryer, T. D., O’Brien, J. T., Rowe, J. B., Menon, D. K., Bor, D., \\& Stamatakis, E. A. (2022). A synergistic core for human brain evolution and cognition. Nature Neuroscience, 1–12. https://doi.org/10.1038/s41593-022-01070-0

 Mediano, P. A. M., Rosas, F. E., Luppi, A. I., Jensen, H. J., Seth, A. K., Barrett, A. B., Carhart-Harris, R. L., \\& Bor, D. (2022). Greater than the parts: A review of the information decomposition approach to causal emergence. Philosophical Transactions of the Royal Society A: Mathematical, Physical and Engineering Sciences, 380(2227), 20210246. https://doi.org/10.1098/rsta.2021.0246

 And was also presented at the Organization for Computational Neuroscience flagship conference:

 \\url{https://cns2020online.sched.com/event/cuJH/w1-s12-multi-target-information-decomposition-and-applications-to-integrated-information-theory}

 \\end{quote} 

 While I was not privy to the peer-review process of those publications (which span a number of different journals with different scopes, including another PLoS journal), the $\\Phi$ID framework clearly passed muster for those reviewers. It is not currently quite as unknown as perhaps it was previously.

 However, I do think that the points the reviewer made \\textit{are} important ones that I have never seem addressed in the literature. We have added the following:

 \\begin{quote}

 The $\\Phi$ID framework deviates from the PID framework in one key way. In the original formulation by Williams and Beer, the lattice (which motivates the Mobius inversion) is derived from the axiomatic properties of the proposed redundancy measure. While there has never been universal agreement on the specific definition of ``redundancy", any function that satisfies the original axioms can be shown to induce the antichain lattice. The lattice follows from the definition of redundancy. In contrast, in the $\\Phi$ID framework, the double-redundancy lattice is not derived from the properties of the $I_{\\cap}^{\\boldmath{\\alpha}\\to\\boldmath{\\beta}}$ function, but rather, imposed by the decision to take the product of the forward and reveres PI lattices. To address this, Mediano et al., imposed a compatibility on any double-redundancy function \\cite{mediano_beyond_2019,mediano_towards_2021}. Given two (potentially, but not necessarily) multivariate) variables \\textbf{X}, \\textbf{Y} and two double-redundancy atoms $\\boldmath{\\alpha}, \\boldmath{\\beta} \\in \\mathcal{A}^{2}$

 \\begin{equation}

 I_{\\cap}^{\\boldmath{\\alpha}\\to\\boldmath{\\beta}} = 

 \\begin{cases}

 I_{\\cap}(\\textbf{X}^{\\alpha_1},\\ldots,\\textbf{X}^{\\alpha_k} ; \\textbf{Y}^{\\beta_1}) & \\iff |\\textbf{Y}|=1 \\nonumber \\\\

 I_{\\cap}(\\textbf{Y}^{\\beta_1},\\ldots,\\textbf{Y}^{\\beta_j} ; \\textbf{X}^{\\alpha_1}) & \\iff |\\textbf{X}|=1 \\nonumber \\\\ 

 I(\\textbf{X};\\textbf{Y}) & \\iff |\\textbf{X}| = |\\textbf{Y}| = 1

 \\end{cases}

 \\end{equation}

 The compatibility axiom requires that, if one of the variables (\\textbf{X} or \\textbf{Y}) is univariate, then the double redundancy function reduces to a classic, single-target redundancy function, and the $\\Phi$ID reduces to the classic PID.

 \\end{quote}

 And later, with respect to the $I_{\\tau sx}$ function specifically:

 \\begin{quote}

 From the definition of $i_{\\tau sx}$ it is clear that it follows the compatibility criteria proposed by Mediano et al., \\cite{mediano_beyond_2019,mediano_towards_2021}. If we are decomposing $i(\\textbf{x},\\textbf{y})$ and $|\\textbf{y}|=1$, then Eq. \\ref{eq:tausx} is equivalent to Eq. \\ref{eq:isx} as the union of all sources ($\\textbf{b}^1\\cup\\ldots\\cup\\textbf{b}^m$) is equivalent to the single source $y$, and likewise for the condition where $|\\textbf{x}|=1$. This shows that $i_{\\tau sx}$ is consistent with both the forward and backwards PIDs (inducing the standard single-target redundancy lattice).

 \\end{quote}

 The distinction between the axiomatic approach of Williams and Beer versus the pre-computed product lattice of Mediano and Rosas is a profound one. As it stands, it is beyond the scope of this project (and likely, my ability) to replicate the derivation that Williams did right at this time. Given the apparent success of the $\\Phi$ID framework in getting published recently, I think it is acceptable to focus on it. However, the reviewer \\textit{has} convinced me that $\\Phi$ID may still be refined and placed on firmer mathematical footing. We leave this for future work, however.

 \\color{black}

 5. Following on from the prior point, you would also need to show that the double-redundancy

 function that you define satisfies these axioms. If you can't (i) show that the double-redundancy

 lattice can be derived from the axioms and then (ii) show that your double-redundancy function

 satisfies these axioms, then you have no justification for later some of the later steps such as

 the Moebius inversion. You will notice that almost all of the existing proposals for a measure

 of redundancy for PID first start out by showing that the proposed measure satisfies the Williams

 and Beer Axioms, or variants thereof.

 \\response{See our response to Point 4, which covers this issue as well. We have added a discussion of how $I_{\\tau sx}$ satisfies the compatibility axiom of Mediano et al., and by extension, satisfies the single-target Williams and Beer axioms (when appropriate).}

 6. Since you are building so directly upon $i_{sx}$, I think you need to improved the description of the

 motivation behind its introduction. As it is, there isn't much beyond the sentence appearing

 before equation (15). There is a great deal of discussion motivating $i_{sx}$ in the original

 papers, so it should be expanded since you are basically just extending this approach to make it

 compatible with Phi-ID.

 \\response{We have added the following to the introduction of $i_{sx}$:}

 \\color{blue}\\begin{quote}

 The classic, local mutual information is \\textit{bivariate}, quantifying the information shared between two variables. To construct a redundancy function that accounted for multiple sources redundantly disclosing information about a single target, Makkeh et al leveraged a link between redundant information and logical implication \\cite{gutknecht_bits_2021,makkeh_introducing_2021}. Briefly, given some set of logical statements $\\psi^1,\\ldots,\\psi^k$, the information that is \\textit{redundantly} disclosed by all of them is the information learned if $\\psi^1$ is true OR $\\psi^2$ is true OR $\\ldots$ $\\psi^k$ is true. From this, they define a logical redundancy measure that induces the same antichain lattice as the PID, with a mapping between every $\\boldmath{\\alpha}\\in\\mathcal{A}$ and a logical statement. For example, the atom $\\{1\\}\\{2\\}\\{3\\}$ maps to the statement $\\psi^1 \\lor \\psi^2 \\lor \\psi^3=True$, and the atom $\\{1\\}\\{2,3\\}$ maps to the statement $\\psi^1 \\lor (\\psi^2\\land \\psi^3)=True$ and so on. The application to random variables is straightforward: given some set of source variables informing on a target \\textit{y}, the information about $y$ redundantly disclosed by all the sources is the information that could be learned by observing $x^1$ alone OR $x^2$ alone $\\ldots$ OR $x^k$ alone. The logic extends to more complicated atoms, such as $\\{1\\}\\{2,3\\}$, which is the information about $y$ that would be learned by observing just $x^1$ alone OR the joint state of $x^2$ AND $x^3$ together. 

 As in the case of local mutual information, $i_{sx}$ defines ``disclosing information" in terms of probability mass exclusions. For example, observing $X^1=x^1 \\lor (X^2=x^2\\land X^3=x^3)$ changes the probability of observing $Y=y$. As with the local mutual information, depending on how $P(y)$ changes, the value of $i_{sx}$ can be positive or negative.

 \\end{quote}

 \\color{black}

 7. Similarly, when you introduce $I_{\\tau sx}$, you do not provide much of a description of the

 motivation behind this particular definition. In particular, you should be justifying the

 replacement that you make in your definition relative to the original definition of $i_{sx}$ provided

 in equation (15). Furthermore, is this change obvious? I am not saying that it is, but since the

 original $i_{sx}$ was proposed for PID rather than Phi-ID, then $i_{sx}$ would clearly also need to be

 modified. Perhaps the change you have made it a trivial change, or maybe it is more significant

 in some way. Either case is fine, but you really aught to be discussing this change and

 justifying why you have defined this generalisation in this particular way.

 \\response{We have added the following:}

 \\color{blue} 

 \\begin{quote}

 It is worth considering the intuition behind this change. Suppose $\\textbf{x}=\\{x^1,x^2\\}$ and $\\textbf{y}=\\{y^1,y^2\\}$. We are interested in the double redundancy $i_{\\tau sx}(x^1,x^2;y^1,y^2)$: the information about $y^1$ OR $y^2$ that could be learned by observing $x^1$ OR $x^2$. There are some global configurations of \\{\\textbf{x},\\textbf{y}\\} that are be consistent with either $y^1$ OR $y^2$ (given by $P(y^1 \\cup y^2)$). If we exclude all of those past-future configurations that are also \\textit{not} consistent with $x^1$ OR $x^2$ (i.e. \\textit{are} consistent with $\\bar{x}^1\\cap\\bar{x}^2$), the double-redundancy is a function of the relative change in $P(y^1\\cup y^2)$. Formally:

 \\end{quote}

 As for whether this is a trivial or significant, I don't really feel like I can rule on that one way or another. It's certainly not a tremendous change to the function, but it also (in my opinion) opens a lot of doors that ``vanilla" $i_{sx}$ could not.

 \\color{black}

 8. You are proposing a decomposition that is based on the local mutual information, which is a non-negative function. The justification for doing this seems to come from reference [35] which gets around this issue by decomposing the local entropy instead of the local mutual information, i.e. the local mutual information is first decomposed into two non-negative local entropies, and then each of these local entropies are decomposed separately. Thus, really you have two separate information decompositions (i.e. two separate decompositions of the local entropy), and you are combining the corresponding atoms of these two separate decompositions quantities to get local-mutual-information-like quantities. In other words, you are not really decomposition the local mutual information, but rather you are decomposing the two local entropies.

 You present your decomposition as though it is decomposition of the local mutual information, and make it sound like the decomposition of the local entropies is an additional feature. To me, it seems like it is actually a decomposition of the local entropy, and you are also combining corresponding atoms of these two separate decompositions quantities to get local-mutual-information-like quantities. This difference is important as the results from the recombine atoms really should be seen as the results obtained from combining quantities from two separate decompositions, rather than a decomposition in its own right. This problem is also present in reference [10].

 \\response{In this regard, I was following the pattern originally set by Makkeh et al., when they introduced the $I_{sx}$ measure, which in turn followed from the work by Connor Finn in the work published from his PhD thesis. There, he established the convention of referring to the entropic components of local mutual information as $i^+$ and $i^-$ respectively. For consistency with prior literature, I would like to continue with this pattern. 

 We have revisited the notion of the partial entropy decomposition, and re-written a large section, adding the following:

 \\begin{quote}

 The full $i_{\\tau sx}^{\\boldalpha\\to\\boldbeta}$ function does not satisfy the partial ordering (monotonicity) criteria, however. Like $i_{sx}$, it can be negative or positive, depending on the structure of the dependency between the elements. The double-redundancy \\textit{can} be decomposed, however, into three \\textit{redundant entropy terms} that are non-negative, partially ordered, and consistent with the integrated information lattice. These three redundant entropy terms induce three partial entropy decompositions \\cite{ince_partial_2017,finn_generalised_2020}: two marginal decompositions on the classic redundancy lattice, and a joint decomposition on the product lattice. 

 \\subsubsection{Multi-Target Redundancy \\& Entropy Decomposition}

 The double redundancy function $i_{\\tau sx}^{\\boldalpha\\to\\boldbeta}$ can be re-written in terms of sums and differences of union entropies. We can re-write Eq. \\ref{eq:tausx} in an equivalent form:

 \\begin{align}

 i_{\\tau sx}&(\\boldsymbol{\\alpha} \\to \\boldsymbol{\\beta}) = \\\\ &\\log_2\\frac{1}{P(\\textbf{a}^1 \\cup\\ldots\\cup\\textbf{a}^k)} \\nonumber \\\\ 

 + &\\log_2\\frac{1}{P(\\textbf{b}^1 \\cup\\ldots\\cup\\textbf{b}^m)} \\nonumber \\\\

 - &\\log_2\\frac{1}{P((\\textbf{a}^1 \\cup\\ldots\\cup\\textbf{a}^k)\\cap(\\textbf{b}^1 \\cup\\ldots\\cup\\textbf{b}^m))} \\nonumber

 \\end{align}

 For proof of equivalence, see Appendix A. 

 For some multivariate \\textbf{x}, recall that $i(\\textbf{x};\\textbf{x})=h(\\textbf{x})$. We can decompose the joint entropy $h(\\textbf{x})$ by assessing how different combinations of \\textit{parts} (i.e. all $x^i\\in\\textbf{x}$) redundantly and synergistically disclose information about the \\textit{whole} \\cite{makkeh_introducing_2021}. If we use the single-target $i_{sx}$ function to decompose $i(x^1,\\ldots,x^k;\\textbf{x})$, we will find that it is equal to $i^+(x^1,\\ldots,x^k;\\textbf{x})$: there is no misinformative component (for proof, see Appendix B).

 Intuitively, we can understand the redundant entropy function as quantifying how much uncertainty about $\\textbf{x}$ is resolved by learning $x^1\\lor\\ldots\\lor x^k$. We will denote the this as the redundant entropy function $h_{sx}$, after \\cite{makkeh_introducing_2021}:

 \\begin{equation}

 h_{sx}(\\boldalpha) = \\log_2\\frac{1}{P(\\textbf{a}^1 \\cup\\ldots\\cup\\textbf{a}^k)}

 \\end{equation}

 Which has been previously shown to satisfy the relevant Williams and Beer axioms locally \\cite{makkeh_introducing_2021}. We can then re-write Eq. \\ref{eq:tausx} as:

 \\begin{equation}

 \\label{eq:ped}

 i_{\\tau sx}(\\boldalpha\\to\\boldbeta) = h_{sx}(\\boldalpha) + h_{sx}(\\boldbeta) - h_{sx}(\\boldalpha\\cap\\boldbeta)

 \\end{equation}

 This framing provides a different, but complementary perspective on $i_{\\tau sx}$. The first term, $h_{sx}(\\boldalpha)$ quantifies how uncertainty about the global joint state (\\textbf{x},\\textbf{y}) is resolved by learning past states $\\textbf{a}^1$ OR $\\dots\\textbf{a}^k$ etc. Similarly, $h_{sx}(\\boldbeta)$ quantifies the uncertainty about (\\textbf{x},\\textbf{y}) resolved by learning future states $\\textbf{b}^1$ OR$\\ldots\\textbf{b}^m$.

 The final term, $h_{sx}(\\boldalpha\\cap\\boldbeta)$ is a little bit less straightforward, and reflects the structure of the double redundancy lattice. We can show that it satisfies the required partial ordering imposed by Mediano et al., \\cite{mediano_beyond_2019}. From \\cite{makkeh_introducing_2021}, we know that, if $\\boldalpha\\preceq\\boldalpha'$ on the marginal (classic) redundancy lattice, then the set of configurations consistent with $\\boldalpha'$ is a subset of those configurations consistent with $\\boldalpha$. This ensures that $h_{sx}(\\boldalpha) \\leq h_{sx}(\\boldalpha')$, and likewise for $\\boldbeta.$ The double redundant entropy term $h_{sx}(\\boldalpha\\cap\\boldbeta)$ quantifies the probability of the \\textit{intersection} of the configurations consistent with $\\boldalpha$ AND $\\boldbeta$. If $\\boldalpha'\\subseteq\\boldalpha$ and $\\boldbeta'\\subseteq\\boldbeta$, then $\\boldalpha'\\cap\\boldbeta'\\subseteq\\boldalpha\\cap\\boldbeta$ and consequently, $h_{sx}(\\boldalpha\\cap\\boldbeta) \\leq h_{sx}(\\boldalpha'\\cap\\boldbeta')$. For a worked example, see Appendix C.

 From Eq. \\ref{eq:ped} we can also construct the informative and misinformative probability mass exclusion formulation of $i_{\\tau sx}$ equivalent to the informative and misinformative components of $i_{sx}$, however unlike in the single-target case, $i_{\\tau sx}^-$ does not follow the partial ordering criteria that $h_{sx}$ does and is also not strictly non-negative. 

 \\end{quote} }

 9. The synthetic systems that you use here in this paper: where do they come from? Are you

 introducing them for the first time here, or are they well known within the field of IIT? Do we

 know what the actual results should be or are their certain results which are seen as desirable

 based on intuition?

 \\response{The systems were ``designed" by me to explore the limit cases of $\\Phi^{WMS}$'s behavior: one where the whole is trivially reducible to the parts (the disintegrated system), one where the whole was entirely irreducible to the parts (the integrated system), and one that showed a mix of the two. These are, to the best of my knowledge, not canonical systems, although they're simple enough that any readers familiar with PID or IIT should get the gist of them. We have added the following:}

 \\color{blue}

 \\begin{quote}

 ``These systems were designed to show the two extreme behaviours of $\\Phi^{WMS}$: the first system is totally dis-integrated and $\\Phi^{WMS}=0$ bit. The second system is completely integrated: the sum of the excess entropy of the parts is 0 bit, while the whole is has non-zero excess entropy. The third system is a heterogeneous combination of integrated and dis-integrated dynamics. Considering the limiting cases of $\\Phi^{WMS}$, we can build intuition about how the $\\Phi ID$ framework with $I_{\\tau sx}$ describes diverse kinds of systems. We hypothesized that the disintegrated system should, generally, not have much redundant temporal mutual information, as the elements are independent of each-other, and there should be little information transfer between individual elements. Similarly, in the case of the integrated system, we expected low redundancy, and a high degree of synergy (as the future of the whole can only be partially predicted by knowing the past of the whole)."

 \\end{quote}

 In general, going into the toy models, we did \\textbf{not} know exactly what the results should be (although they could be done by hand). This is part of what makes them an interesting starting point, however: they can demonstrate where the $\\Phi ID$ framework diverges from our intuitions, and are simple enough that those divergences might be explainable.\\color{black}\\newline

 In general, your description of obtain results does not give me the impression that the results are well understood. For instance, when it comes to describing the result for the disintegrated system, you describe the the element-wise storage atoms as being the "strongest" while the pairwise transfer atoms are negative. You describe this latter result as being "consistent with the notion of the system as a disintegrated structure", but I don't see how this is the case and

 you do not provide any further clarification. You say that there are more positive interactions(?) between individual elements and higher-order synergistic joint states than anticipated. However, you never described or justified your anticipated result, so this is not a

 helpful comment to make. The fact that you describe this result a "curious" does not fill me with confidence that the results are indeed sensible or helpful. These results are all local mutual informations, but as described in point 7, these values are really obtained from separate decompositions of the two local entropies, so I suspect these results would be less mysterious if

 you looked at these values.

 Similarly, you are also a bit vague and uncertain about the results obtained from the integrated system. Do we know what the results should be, or are is there some kind of intuitive result that you would like to see? Describing some of the obtained results as a "surprise" when you are considering synthetic examples that should have some definite result is quite problematic to me. This could be fine if you provide an explanation for the surprising results, but I don't see any justification that explains these results to me. This kind of description would be absolutely fine if this was a well established method and you were applying it to some empirical

 data from some system that is not well understood. It is troubling, however, when these are the results that you obtain from applying this to synthetic examples which really should be justifying the theoretical definition that you have introduced.

 Regarding the heterogeneous system, I am not really sure what we are supposed to take away from this example. You were actually a bit more explicit in stating your expectation for this example, in that you said that you expected to see multiple types of information atoms present, and indeed, this is the result that you obtained. However, what does it really tell us to learn that a randomly generated example produces a dynamic which is associated with a small amount of

 information associated with each of the atoms? Without further guidance, I am not sure what exactly I should take away from this.

 \\response{There are a few points to respond to here. The first is that I had initially tried to err on the side of caution when it came to interpreting the results (which perhaps came off as waffling). My concern was ``over interpreting" what I saw. However, I have re-written the relevant sections to more emphatically reflect my interpretations.

 \\begin{quote}

 Each of the three synthetic system is comprised of two, binary, elements that evolve through times according to different Markovian state-transition networks (visualized in Fig. \\ref{fig:synth_systems}). Prior work on such simple, Boolean networks has shown that the space of even very small systems has a surprisingly rich distributions of redundant, unique, and synergistic effective information atoms \\cite{varley_emergence_2022}. Despite the simplicity of the synthetic systems under study here, we can see how $I_{\\tau sx}$ can reveal markedly different dynamic regimes. These systems were designed to show the two extreme behaviours of $\\Phi^{WMS}$: the first system is totally dis-integrated and $\\Phi^{WMS}=0$ bit (as the future of the whole can be perfectly predicted from the independent parts). The second system is completely integrated: the sum of the excess entropy of the parts is 0 bit, while the whole is has non-zero excess entropy. The third system is a heterogeneous combination of integrated and dis-integrated dynamics. Considering the limiting cases of $\\Phi^{WMS}$, we can build intuition about how the $\\Phi$ID framework with $I_{\\tau sx}$ describes diverse kinds of systems. We hypothesized that the disintegrated system should, generally, not have much redundant temporal mutual information, as the elements are independent of each-other, and there should be little information transfer between individual elements. Similarly, in the case of the integrated system, we expected low redundancy, and a high degree of synergy (as the future of the whole can only be partially predicted by knowing the past of the whole). 

 \\subsubsection{Disintegrated System}

 The first system, $\\textbf{S}^D$ is a ``disintegrated" system, in that each of the two dynamic elements is disconnected from the other: both predict \\textit{their own} futures with total determinism (the pattern is an oscillation $1\\to0\\to1\\to\\ldots$), however there is no integration. Consequently, the excess entropy $I(\\textbf{S}^D_{t-1} ; \\textbf{S}^D_{t}) = 2$ bit, and both individual excess entropies are each 1 bit: the ``whole" is trivially reducible to the sum of its parts, since there's no actual interaction between elements. For a visualization of the state-transition matrix, see Figure \\ref{fig:synth_systems}, left.

 Decomposing the excess entropy using $I_{\\tau sx}$ reveals several interesting relationships (for the full decomposition, see Table \\ref{tab:synth_systems}). As expected, the strongest information atoms are the element-wise information ``storage" atoms: $\\{1\\}\\to\\{1\\}$ and $\\{2\\}\\to\\{2\\}$ atoms. This is consistent with our intuition that the future of each element is \\textit{best} predicted by its own immediate past. The fact that the unique information storage atoms are also the largest single atoms is consistent with the idea that the most informative information dynamic about the whole system is the behaviour of the individual nodes considered individually. The informative interaction between the unique information and the synergistic information are also consistent with our intuitions about the disintegrated systems. For instance, the informative value of the ``upward-causation" atom $\\{1\\}\\to\\{12\\}$ reflects the fact that knowing the state of $X^1_{-\\tau}$ also informs on the state of the whole $\\textbf{X}_{t}$ in the future: knowing $X^1_{-\\tau}=0$ rules out any configuration of $\\textbf{X}_{t}$ where $X^t=0$ (since the values of each $X^i$ oscillate). Likewise for the ``downward-causation" atoms such as $\\{12\\}\\to\\{1\\}$: knowing the state of the \\textit{whole} at time $t-\\tau$ constrains the individual parts at time $t$ (albeit, to a lesser extent than they constrain themselves).

 The non-zero value of the double redundancy atom $\\{1\\}\\{2\\}\\to\\{1\\}\\{2\\}$ is unexpected, although not inexplicable. Suppose that, at time $t-\\tau$ we learn that $X^1_{-\\tau}=0$ OR $X^2_{-\\tau}=0$. Since both $X^i$ oscillate $0\\to1\\to0\\to1\\ldots$, learning the state of either past variable is enough to rule out one possible joint future: $\\textbf{X}_t=(0,0)$. Consequently, the union probability of all possible futures consistent with $X^1_t\\not=0$ OR $X^2_t\\not=0$ increases. 

 The final set of atoms worth exploring are the negative value for the information transfer terms (such as $\\{1\\}\\to\\{2\\}$). The temporal mutual information $I(X^1_{-\\tau};X^2_t)=0$ bit, but the partial information atom is less than zero. Why? The answer is that computing the information transfer atom requires subtracting the sum of all the atoms that precede it on the lattice from the temporal mutual information. In this case, this is only the non-zero, double redundancy atom $\\{1\\}\\{2\\}\\to\\{1\\}\\{2\\}$, which when subtracted off, produces a negative value (since the temporal mutual information is 0 bit). 

 \\subsubsection{Integrated System}

 The second system, $\\textbf{S}^G$ is an ``integrated" system, in that there the whole system has 1 bit of excess entropy, but both elements have individual excesses entropies of 0 bit. This is accomplished using a parity check function: at every time step, the parity of the system is preserved, but the individual assignments are done randomly. For example, if $\\textbf{S}^{G}_{-\\tau} = (1, 0)$, then $\\textbf{S}^{G}_{t}$ could equal $(0,1)$ or $(1,0)$ with equal probability (but never $(0,0)$ or $(1,1)$). For a visualization of the state-transition matrix, see Figure \\ref{fig:synth_systems}, center.

 The first interesting finding is that, as with $\\textbf{S}^D$, the double-redundancy atom is unexpectedly positive. This occurs because, as before, learning $X^1_{-\\tau}$ OR $X^2_{-\\tau}$ is sufficient to inform on the future. Suppose we learn that $X^1_{-\\tau}=0$ OR $X^2_{-\\tau}=0$. There are two configurations with an odd-parity consistent with those conditions ((0,1) and (1,0)), and only one configuration with even parity ((0,0)). From this, we are lead to believe that the future state will also be more likely to have an odd parity than an even one. Due to the synergistic nature of the parity-check function, we cannot directly project from the parity of the joint state back down onto the states of the individual elements, which explains the negative values of the various information copy and erase atoms (such as $\\{1\\}\\to\\{1\\}\\{2\\}$ and $\\{1\\}\\{2\\}\\to\\{1\\}$). The non-negative values of the various unique information storage and transfer atoms ($\\{1\\}\\to\\{1\\}$ and $\\{1\\}\\to\\{2\\}$) occurs because the mutual informations between any pair of elements are all 0 bit (i.e. $I(X^1_{-\\tau};X^1_t)=0$ bit, $I(X^1_{-\\tau};X^2_t)=0$ bit, etc), however, the sum of all atoms lower down on the lattice is negative. For example, consider atom $\\{1\\}\\to\\{1\\}$. The excess entropy $E(X^1) = 0$ bit, however $\\{1\\}\\{2\\}\\to\\{1\\}\\{2\\} + \\{1\\}\\{2\\}\\to\\{1\\} + \\{1\\}\\to\\{1\\}\\{2\\} = -0.152$ bit, so the value of $\\{1\\}\\to\\{1\\}$ works out to $0 - (-0.152)$ bit: 0.152. The result can be admittedly difficult to interpret, although negative atoms in classic, single-target PIDs are also relatively widespread (see \\cite{ince_measuring_2017,finn_pointwise_2018,makkeh_introducing_2021}). One possible interpretation, explored more in Section \\ref{sec:negative} is that these values reflect how the \\textit{context} provided by the dynamics of the whole system can influence our interpretation of dynamics of the individual parts. For example, the when considered alone, there is no predictive information about the future of $X^1$ in it's own past, however, $X^1$'s evolution through time is not an autonomous process, but occurs \\textit{in the context of} $X^2$'s dynamics. Consequently, the kinds of inferences we can make about $X^1$ when we consider the other elements of the system may be different from the kinds of inferences we would make if we were considering $X^1$ alone.

 The higher-order, synergistic atoms show the same effect: since the redundancy-to-synergy atoms ($\\{1\\}\\{2\\}\\to\\{12\\}$ and $\\{12\\}\\to\\{1\\}\\{2\\}$) are both negative, the upward and downward-causation atoms are positive, despite the fact that the relevant mutual informations are all zero bit. 

 \\subsubsection{Heterogeneous System}

 The final system was one with heterogeneous transitions, with probabilities drawn from a Gaussian distribution $\\mathcal{N}(0,1)$ (for details, see Varley \\& Hoel \\cite{varley_emergence_2022}). In contrast to the prior two systems, this system, $\\textbf{S}^{H}$ does not have an \\textit{a priori} fixed ``type" of dynamic and was expected to display multiple types of information conversion. From the outset, we anticipated evidence of synergistic dynamics, as the excess entropy of the whole system was 0.422 bit, while each of the two elements had individual temporal mutual informations of 0.017 bit and 0.001 bit respectively, indicating a dynamic where the whole is much more predictive than the sum of its parts. For a visualization of the state-transition matrix, see Figure \\ref{fig:synth_systems}, right.

 Consistent with expectations, $\\textbf{S}^H$ did not have the same regularity of information dynamics displayed by $\\textbf{S}^D$ and $\\textbf{S}^G$: for example, the atom $\\{1\\}\\{2\\} \\to \\{1\\}$ was negative, indicating a misinformative relationship, while $\\{1\\}\\{2\\} \\to \\{2\\}$ was positive and had a greater absolute value. Similarly, the conversion from redundant to synergistic information and vice versa both has opposite signs, suggesting that this system simultaneously displays informative ``downward causation", but misinformative ``upward causation". In totality, there were more informative integrated information atoms than misinformative ones (a ratio of 11 to 5), showing that, despite the overall strongly synergistic nature of the system, unique information transfer and redundant information dynamics all co-existed together. This is consistent with previous work that found that these kinds of modified-Gaussian systems can display a wide range of information dynamics, at multiple scales \\cite{varley_emergence_2022}.

 \\subsubsection{Interpreting Negative Local $\\Phi$I Atoms}

 \\label{sec:negative}

 The three systems described above provide concrete toy models which we can use to build intuition about the phenomena of negative $\\Phi$I values. Consider the heterogeneous system $\\textbf{S}^{H}$: specifically the information that $S^{2}_{t-1}$ communicates to itself at time $t$ ($I(S^{2}_{t-1} ;S^{2}_{t})$). This value, sometimes called active information storage \\cite{lizier_local_2013}, is an expected mutual information and must be non-negative: in this case, it is 0.001 bit. If an observer were just observing the dynamics of $S^{2}$ (and ignoring $S^{1}$ entirely), their uncertainty about the future of $S^{2}$ would be reduced by observing its past. Despite this, the ``stored" partial information $\\{2\\}\\{2\\}$ is negative. How can this occur?

 Our interpretation is that this mismatch is explained by the fact that the evolution of $S^{2}$ occurs \\textit{in the context of all other elements in the system.} Analysing $S^{2}$ on its own can be misleading, because its dynamics are informed by the states of the other elements. In this case, the lions share of the information is information that initially present in both atoms, and then erased from $S^{1}$: $\\{1\\}\\{2\\}\\to\\{2\\}$. So, a significant amount of the information that one observes in $I(S^{2}_{t-1};S^{2}_t)$ is not \\textit{specific} to $S^{2}$ (at least not at first), but rather, emerges from the interaction with $S^{1}$.

 In the case of these toy systems, there is no ``mechanism" to be explored, however, this kind of distinction may be of value when analysing real-world systems. For example, a scientist studying the activity of neurons may observe a non-zero active information storage and propose to connect that to biophysical processes such as refractory periods \\cite{varley_information_2022}, however, the $\\Phi$ID analysis shows that this information may not actually reflect the specific dynamics of a single neuron, but rather, some contextual interaction between two, and the \\textit{actual} memory capacity of the neuron is altogether different (in this case, misinformative, rather than informative). This example is, at present, admittedly speculative, considerable work remains to understand how the outputs of the $\\Phi$ID algorithm map onto complex, real-world dynamical process. 

 \\end{quote}

 }

 10. The empirical example using a dissociated neural culture: I don't want to be overly negative

 here because I think the analysis of this kind of data is an interesting scientific endeavour,

 and I think that information-theoretic methodologies are an appropriate way of potentially

 understanding this kind of system. However, this entire section to me, is not helpful for a

 paper that aims to propose a novel theoretical approach. It is interesting, but I think that

 this work would be far more appropriate as a separate paper after this theoretical method has

 been accepted in its own right.

 The theoretical result should be interesting based on its own merits. You should propose the

 theoretical measure, provide the motivation for the definition, and justify it using

 well-understood examples. Typically, for this kind of proposal, you would typically use

 synthetic examples (indeed, as you have aimed to do with your synthetic examples). This is

 certainly true for all of the PID based approaches, although perhaps it is more tricky for

 Phi-ID (see point 1).

 The problem here is that the motivation for defining your measure is not very clear (see points

 6 and 7), and I think there are some notable theoretical shortcomings (see points 4, 5 and 8).

 Then the results the you obtain for the synthetic examples are not clear or convincing (see

 point 9). With all of this in mind, I do not see what value the empirical example where there

 is no well-established ground truth. If the results can't be fully explained for the synthetic

 examples that are designed to exhibit certain features or characteristics, then I don't see how

 we can trust the results when applied to real empirical data.

 On the flip side, it is you paper, so I don't want to say that this analysis must be removed.

 Rather, I am saying that it needs much better theoretical justification before we can trust the

 results, or draw inference from the results when applied to empirical data. To reiterate, the

 theoretical results should be interesting in there own right, and should stand on their own two

 feet (to use an idiom). In my opinion, the level of detail required achieve this level of

 theoretical rigour would result in a paper in its own right, and then this example could

 potentially go in a separate empirical paper that uses the newly establish theoretical

 framework. Whether or not you want to keep the empirical example in this paper, you must

 improve the rigour of the theoretical results before this paper is suitable for publication.

 \\response{I feel quite strongly that the empirical results should be kept. I think it is important that there be at least \\textit{some} application to a real-world system to show that this is not simply a theoretical exercise. This was inspired by the original introduction of $\\Phi$ID by Mediano et al., who worked through the derivation of the double-redundancy lattice, and then ended with a small number of ``real-world" applications, including an analysis of invasive ECoG recordings from macaques.

 There, as in here, the goal was not to claim a ground-breaking new result, but rather, to provide a kind of proof-of-concept that could inspire interested researchers and demonstrates how the analysis could be applied. The best of our knowledge, this kind of frame-wise, local information-theoretic analysis of avalanche profiles has never been done in neural cultures. The hope is that the work presented here will motivate future work that \\textit{can} dig into the gory detail more. In this context, the fact that the results are largely descriptive is not necessarily a deal-breaker. }

 \\bibliographystyle{unsrt}

 \\bibliography{temporal_redundancy.bib}

\\end{document}

---

## [Decision Letter · Decision Letter 1]

28 Feb 2023

Decomposing past and future: Integrated information decomposition based on shared probability mass exclusions

PONE-D-22-09166R1

Dear Thomas Varley

,

We’re pleased to inform you that your manuscript has been judged scientifically suitable for publication and will be formally accepted for publication once it meets all outstanding technical requirements.

Kind regards,

Benjamin Z. Webb, Ph.D.

Academic Editor

PLOS ONE

P.S. There are just a few comments by the second referee that could be addressed in a slightly revised version of the paper but whether these changes are placed in the final version of the paper is up to the author.

Reviewers' comments:

Reviewer's Responses to Questions

Comments to the Author

1. If the authors have adequately addressed your comments raised in a previous round of review and you feel that this manuscript is now acceptable for publication, you may indicate that here to bypass the “Comments to the Author” section, enter your conflict of interest statement in the “Confidential to Editor” section, and submit your "Accept" recommendation.

Reviewer #2: (No Response)

Reviewer #3: All comments have been addressed

2. Is the manuscript technically sound, and do the data support the conclusions?

Reviewer #2: Yes

Reviewer #3: Yes

3. Has the statistical analysis been performed appropriately and rigorously? 

Reviewer #2: Yes

Reviewer #3: (No Response)

4. Have the authors made all data underlying the findings in their manuscript fully available?

Reviewer #2: Yes

Reviewer #3: Yes

5. Is the manuscript presented in an intelligible fashion and written in standard English?

Reviewer #2: Yes

Reviewer #3: Yes

6. Review Comments to the Author

Reviewer #2: * Summary

The author has greatly improved the manuscript, and my recommendation is that the paper can go to publication pending minor revisions. The author has addressed the 10 issues that I previously mentioned insofar as can reasonably demanded. I am still sceptical of the \\Phi-ID framework regarding the specific issues that were raised in the prior round. However, the author cannot be expected to solve all weaknesses or shortcoming in existing research. I would urge the author to have a further think about these issues going forward, but this certainly should not stop the publication of this work.

I have also added one additional point regarding the bar | notation used at a certain point in this paper. However, the fix is straightforward.

* PRIOR ISSUES and COMMENTS

1. Okay, this is interesting to know! However, I am still confused by something here: if you are applying it to random variables (instead of timeseries), then what is the meaning of each of the atoms? For instance, in Fig. 2, you say that {1}{2}->{1} corresponds to information that is redundantly disclosed by X^1 and X^2 at time t-\\tau that is then only uniquely disclosed by X^1 at time t. What does the atom {1}{2}->{1} corresponds to if there is no sense of time in the data? I suppose I am saying that I find your response interesting, and it leaves me with even more questions. I don't feel very strongly that you need to address this point, however. This really is something that aught to be addressed by the authors who introduced \\phi-ID.

2. I am a bit confused by the response. I find it natural to query the composition of the information provided by the past states of each component about the future states of the entire system. For example, in a two component system, we would have a decomposition based on I(X^1_-\\tau; \\bm{X}_t) and I(X^2_-\\tau; \\bm{X}_t), which tells us what the past states of X^1 and X^2 tell us about the future states of the entire system (X^1,X^2) in terms of what the past states of each component uniquely, redundantly and synergistically tell us about the future states of the system. Of course, this does not consider the unique, redundant and synergistic interactions between the system in the future. In order to consider these, we would have to slide our time window along, say, k steps into the future, i.e. by evaluating a decomposition based on I(X^1_-\\tau+k; \\bm{X}_t+k) and I(X^2_-\\tau+k; \\bm{X}_t+k). I don't see why we should expect a quantity based on the time t to tell us about the interactions into the future beyond t.

I can see why it would be nice to have such a quantity. However, that does not mean that such a quantity has to exist. The critique here is strongly related to my fourth point; that is, I find the axiomatic basis of \\phi-ID very weak compared to PID. The PID lattice and hence the atoms are a consequence of some rather simple axioms. In contrast, as far as I can tell, the second \\phi-ID axiom effectively introduces the \\phi-ID lattice as an axiom itself without much justification. As such, I don't find the associated atoms very natural, which is why I say it might be nice to have these quantities, but the \\phi-ID axioms don't provide much of a justification to me.

However, this is more a critique of \\phi-ID itself that the content of this paper. The contribution that you make here is perfectly reasonable (and indeed very good) if I suppose that \\phi-ID is itself theoretical sound, so I would not like this point to block you from publishing this work.

3. The point I was making about the target chain rule is that it could be used to consider the how each time point in the future contributes to each atom of information about the entire future. If we consider the future states over time, i.e. \\bm{X}_t = {\\bm{X}t_1, \\bm{X}t_2, ...}, then a decomposition with a target chain rule could be used to consider how each atom of information at each time point in the future contributes to each of the corresponding atom of information for the entire future. For example, the you could look at what the past states redundantly tell us about the future states R(X^1_-\\tau,X^2_-\\tau; \\bm{X}_t) and consider how each time point t_1, t_2... contributes by looking at R(X^1_-\\tau,X^2_-\\tau; \\bm{X}_{t_1}), R(X^1_-\\tau,X^2_-\\tau; \\bm{X}_{t_2} | \\bm{X}_{t_1}) and R(X^1_-\\tau,X^2_-\\tau; \\bm{X}_{t_3} | \\bm{X}_{t_1}, \\bm{X}_{t_2},), etc, via the chain rule:

R(X^1_-\\tau,X^2_-\\tau; \\bm{X}_{t_1}) =

R(X^1_-\\tau,X^2_-\\tau;\\bm{X}_{t_2} | \\bm{X}_{t_1}) +

R(X^1_-\\tau,X^2_-\\tau; \\bm{X}_{t_3} | \\bm{X}_{t_1}, \\bm{X}_{t_2},) + ...

If you have a target chain rule for the redundancy, then you also have it of the synergy. Thus, for instance, you could use a PID with a chain rule to understand how the (total) synergy in the future is composed of synergies at each particular time point in the future. This to me seems like a natural way of using PID to address the problem that you seem interested in regarding point 2. Similar to before, this is more a critique of the \\phi-ID framework than of your contribution to here, which is perfectly reasonable.

4. This is really the main issue that I have, i.e. the weakness that I see in the \\phi-ID approach. However, as I said above, I think your contribution to the literature is very worthwhile.

I think it is also worth saying I have no issue with basing your research on an unpublished paper. \\phi-ID is certainly getting enough attention to merit further work. However, the fact that many people are using \\phi-ID does not make this weakness in the axiomatic basis go away. Personally, I would be far more worried about fixing the weak foundations before trying to build more work on top of it. However, it is perfectly reasonable for you to do this and I would not like this point to prevent you from publishing your worthwhile findings.

5. Your response to point 4 addresses this point perfectly.

6. Your response addresses this point.

7. I did not mean to say that the contribution is not significant -- it is very worthwhile! I more meant that I think you should comment on how natural it is as a generalisation, and your additional comment addresses this perfectly.

8. The changes you have made fully address this point.

9. The new version is much improved. The motivation behind each of the synthetic systems is greatly improved. Now that I understand the motivation behind them, this section is much stronger to me. I think your earlier caution along with the weaker description of the motivation gave me the sense that the results were not well-understood. This improved description together with the improvement regarding point 8 has much improved this section. The intuition is more clear, and I much prefer this section. I am still a little sceptic of some of the interpretations, but this all comes from my scepticism of the terms in the \\phi-ID lattice rather than anything that is said here.

10. Given the improvements to the section regarding the synthetic examples, my main problem with this section has been resolved. (When the results of the synthetic examples where no clear, this application seemed to be an attempt at justifying the method rather than an example of how it can be used.) I perhaps was not explicit enough in saying that my comment here was not to say that I thought this section must be removed.

* NEW ISSUE

1. The notation that you have used for the unique information is quite problematic. Specifically, you have used the bar | to denote which variable the information is unique with respect to, e.g. Unq(X^1;Y|X^2) is the unique information that X^1 provides about Y with respect to X^2. It is standard practice to use the bar | to denote conditional probabilities, and this notation is naturally inherited by information theory as a subset of probability theory. Thus, it is very confusing to use the bar notation for the unique information.

The problem with you choice is that, in probability theory, you can always replace a probability with a conditional probability, and so any function defined based on some probability can also have a conditional variant. So, for instance, instead of decomposing the information that X^1 and X^2 provide about Y which is associated with P(X^1,X^2,Y), as you have denoted in eq. (4), you could decompose the information that X^1 and X^2 provide about Y given that you know variable Z, which is associated with P(X^1,X^2,Y | Z), yielding the decomposition,

I(X^1,X^2,Y|Z) = Red(X^1,X^2;Y | Z) + Unq(X^1; Y \\ X^2 | Z) + Unq(X^2; Y \\ X^1 | Z)

+ Syn(X^1,X^2;Y | Z).

If you use the bar for unique information too, then how could you represent the above decomposition? I think this issue is also in the original \\phi-ID paper, so it probably just came directly from there without noticing this issue. I suggest switching to the set difference \\ notation used by Bertschinger et al 'Quantifying Unique Information', as this is much less problematic, as demonstrated here.

Reviewer #3: (No Response)

7. PLOS authors have the option to publish the peer review history of their article (what does this mean?). If published, this will include your full peer review and any attached files.

Do you want your identity to be public for this peer review?

 For information about this choice, including consent withdrawal, please see our Privacy Policy.

Reviewer #2: No

Reviewer #3: No

---

## [Editor Report · Acceptance letter]

15 Mar 2023

PONE-D-22-09166R1 

Decomposing past and future: Integrated information decomposition based on shared probability mass exclusions. 

Dear Dr. Varley:

I'm pleased to inform you that your manuscript has been deemed suitable for publication in PLOS ONE. Congratulations! Your manuscript is now with our production department. 

Kind regards, 

on behalf of

Dr. Benjamin Z. Webb 

Academic Editor

PLOS ONE